# Near-Optimal Bounds for
# Testing Histogram Distributions

Clément L. Canonne
University of Sydney
clement.canonne@sydney.edu.au

Ilias Diakonikolas*
University of Wisconsin-Madison
ilias@cs.wisc.edu

Daniel M. Kane†
University of California, San Diego
dakane@cs.ucsd.edu

Sihan Liu
University of California, San Diego
sil046@ucsd.edu

## Abstract

We investigate the problem of testing whether a discrete probability distribution over an ordered domain is a histogram on a specified number of bins. One of the most common tools for the succinct approximation of data, $k$-*histograms* over $[n]$ are probability distributions that are piecewise constant over a set of $k$ intervals. Given samples from an unknown distribution $\mathbf{p}$ on $[n]$, we want to distinguish between the cases that $\mathbf{p}$ is a $k$-histogram versus far from any $k$-histogram, in total variation distance. Our main result is a sample near-optimal and computationally efficient algorithm for this testing problem, and a nearly-matching (within logarithmic factors) sample complexity lower bound.

## 1 Introduction

### 1.1 Background and Motivation

A classical approach for the efficient exploration of massive datasets involves the construction of succinct data representations, see, e.g., the survey [CGHJ12]. One of the most useful and commonly used compact representations are *histograms*. For a dataset $S$, whose elements are from the universe $[n] := \{1, \ldots, n\}$, a $k$-histogram is a function that is piecewise constant over $k$ interval pieces. Histograms constitute the oldest and most popular synopsis structure in databases and have been extensively studied in the database community since their introduction in the 1980s [Koo80], see, e.g., [GMP97, JKM+98, CMN98, TGIK02, GGI+02, GKS06, ILR12, ADH+15, Can16], for a partial list of references. In both the statistics and computer science literatures, several methods have been proposed to estimate histogram distributions in a range of natural settings [Sco79, FD81, DL04, LN96, Kle09, CDSS14, ADH+15, ADLS17, DLS18].

In this work, we study the algorithmic task of deciding whether a (potentially very large) dataset $S$ over the domain $[n]$ is a $k$-histogram (i.e., it has a succinct histogram representation with $k$ interval pieces) or is "far" from *any* $k$-histogram representation (in a well-defined technical sense). Our focus is on *sublinear time* algorithms [Rub06]. Instead of reading the entire dataset $S$, which could be highly impractical, one can instead use randomness to sample a small subset of the dataset. Note that

---

*Supported by NSF Medium Award CCF-2107079, NSF Award CCF-1652862 (CAREER), a Sloan Research Fellowship, and a DARPA Learning with Less Labels (LwLL) grant. Some of this work was performed while the author was visiting the Simons Institute for the Theory of Computing.

†Supported by NSF Medium Award CCF-2107547, NSF Award CCF-1553288 (CAREER), and a Sloan Research Fellowship.

36th Conference on Neural Information Processing Systems (NeurIPS 2022).

sampling a (uniformly) random element from $S$ is equivalent to drawing a sample from the underlying probability distribution $\mathbf{p}$ of relative empirical frequencies. This observation brings our algorithmic problem of "histogram testing" in the framework of distribution property testing (statistical hypothesis testing) [BFR$^+$00, BFR$^+$13], see, e.g., [Can20] for a survey.

Formally, we study the following task: for an integer $1 \leq k \leq n$, denote by $\mathcal{H}_k^n$ the set of $k$-histogram distributions over $\{1, 2, \ldots, n\}$, i.e., the set of all distributions $\mathbf{p}$ such that there exists a partition of $[n]$ into $k$ consecutive intervals (not necessarily of the same size) with $\mathbf{p}$ being uniform on each interval. Given access to i.i.d. samples from an unknown distribution $\mathbf{p}$ on $[n]$ and a desired error tolerance $0 < \varepsilon < 1$, we want to correctly distinguish (with high probability) between the cases that $\mathbf{p}$ is a $k$-histogram versus $\varepsilon$-far from any $k$-histogram, in total variation distance (or, equivalently, $\ell_1$-norm). It should be noted that the histogram testing problem studied here is not new. Prior work within the algorithms and database theory community has investigated the complexity of the problem in the past ten years (see, e.g., [ILR12, ADH$^+$15, Can16] and Section 1.4 for a detailed summary of prior work). However, known algorithms for this task are sub-optimal, and in particular there is a polynomial gap between the best known upper and lower bounds on the sample complexity of the problem. At a high level, the difficulty of our histogram testing problem in the sub-linear regime lies in the fact that the location and "length" of the $k$ intervals defining the histogram representation (if one exists) is a priori unknown to the algorithm.

We believe that the histogram testing problem is natural and interesting in its own right. Moreover, a sample-efficient algorithm for this testing task can be used as a key primitive in the context of *model selection*, where the goal is to identify the "most succinct" data representation. Indeed, various algorithms are known for learning $k$-histograms from samples whose sample complexities (and running times) scale proportionally to the succinctness parameter $k$ (and are completely independent of the domain size $n$) [CDSS14, ADH$^+$15, ADLS17]. Combined with an efficient tester for the property of being a $k$-histogram (used to identify the smallest possible value of $k$ such that $\mathbf{p}$ is a $k$-histogram, e.g., via binary search), one can obtain a sketch of the underlying dataset. See Appendix C for a detailed description.

## 1.2 Our Results

Our main contribution is a near-characterization of the sample complexity of the histogram testing problem. Specifically, we provide (1) a sample near-optimal and computationally efficient testing algorithm for the problem, and (2) a nearly-matching sample complexity lower bound (within logarithmic factors). In particular, we establish the following theorem:

**Theorem 1** (Main Result). *There exists a testing algorithm for the class of $k$-histograms on $[n]$ with sample complexity $m = \widetilde{O}(\sqrt{nk}/\varepsilon + k/\varepsilon^2 + \sqrt{n}/\varepsilon^2)$ and running time $\mathrm{poly}(m)$. Moreover, for any $k \in [n]$ and $0 < \varepsilon < 1$, any testing algorithm for the class of $k$-histograms on $[n]$ requires at least $\widetilde{\Omega}(\sqrt{nk}/\varepsilon + k/\varepsilon^2 + \sqrt{n}/\varepsilon^2)$ samples.*

(The $\tilde{O}(\cdot)$ and $\tilde{\Omega}(\cdot)$ notation hides polylogarithmic factors in the argument.) Theorem 1 characterizes the complexity of the histogram testing problem within polylogarithmic factors. Note that there are three terms in the sample complexity; namely, $\sqrt{nk}/\varepsilon$, $k/\varepsilon^2$, and $\sqrt{n}/\varepsilon^2$. The sample complexity of the problem is dominated by one of these three different terms, depending on the relative sizes of $n, k$ and $1/\varepsilon$. An illustration is given in Figure 1.

Prior to our work, the best previous histogram testing algorithm had sample complexity $\widetilde{O}(\sqrt{kn}/\varepsilon^3)$ [CDGR18], while the best known lower bound was $\widetilde{\Omega}(\sqrt{n}/\varepsilon^2 + k/\varepsilon)$ [Can16].[3]

We note that previous upper and lower bounds exhibit a polynomial gap, even for constant values of $\varepsilon$ or $k$. For example, in the "large-$k$" regime where $k = n^c$ for some constant $0 < c < 1$, there was a gap between $\widetilde{O}(n^{1/2+c/2})$ and $\tilde{\Omega}(n^{1/2})$ in the sample complexity. In this regime, however, Theorem 1 results in the near-optimal bound of $\widetilde{\Theta}(n^{1/2+c/2})$. Similarly, in the "high-accuracy" regime where $\varepsilon = 1/n^c$ for some constant $c > 0$ (and, say, constant $k$), previous bounds only established that the sample complexity lies between $\widetilde{O}(n^{1/2+3c})$ and $\widetilde{\Omega}(n^{1/2+2c})$, while our result

---

[3]As discussed in Section 1.4, while an upper bound of $\widetilde{\Omega}(\sqrt{n}/\varepsilon^2 + k/\varepsilon^3)$ is claimed in [Can16], the analysis of their algorithm is flawed; and, indeed, our work shows that the sample complexity bound stated in [Can16] *cannot* hold, as it would contradict our lower bound.

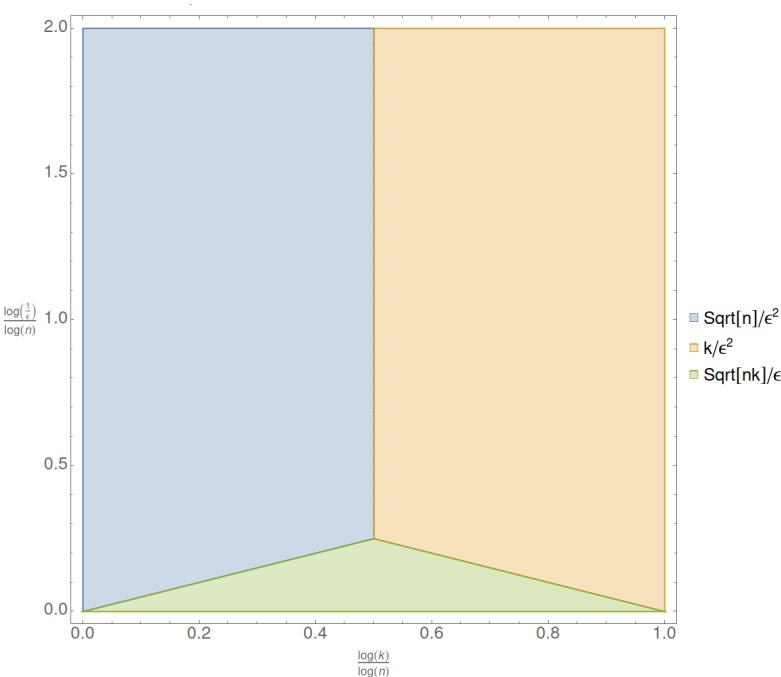

Figure 1: The $x$-axis, $y$-axis are $\log(k)/\log(n)$ and $\log(1/\varepsilon)/\log(n)$ respectively. Each point in the graph corresponds to a setting of $n, k, \varepsilon$, and is colored based on the corresponding dominating term.

shows the (nearly-tight) bound is $\widetilde{\Theta}(n^{1/2+c})$. These are only two specific examples: more generally, the previously known bounds are suboptimal by polynomial factors in $1/\varepsilon$ when $\varepsilon \geq \sqrt{k/n}$; and by polynomial factors in all parameters $k, n, 1/\varepsilon$ when $\varepsilon \leq \sqrt{k/n}$. Theorem 1 settles the sample complexity of the problem, up to logarithmic factors, for *every* parameter setting.

At a technical level, our sample complexity lower bound construction conceptually differs from previous work in distribution testing, drawing instead from sophisticated techniques from the distribution *estimation* literature. Our upper bound follows from the "Testing-via-Learning" framework proposed in [ADK15]. The main technical innovation is a sample- and time- efficient *adaptive* algorithm which can simultaneously learn an unknown histogram *with unknown interval structure* distribution and identify a domain where the learned result is accurate. We elaborate on these aspects next.

## 1.3 Overview of Techniques

**Sample Complexity Lower Bound.** We follow the typical high-level approach in proving sample complexity lower bound. Namely, we define two ensembles of distributions $D_{\text{YES}}$ and $D_{\text{NO}}$ such that, with high probability, the following conditions are satisfied: (1) a random distribution from $D_{\text{YES}}$ is a $k$-histogram, (2) a random distribution from $D_{\text{NO}}$ is $\varepsilon$-far from any $k$-histogram, and (3) given a sample of appropriate size, it is information-theoretically impossible to distinguish a random distribution drawn from $D_{\text{YES}}$ from a random distribution drawn from $D_{\text{NO}}$.

We start by describing our hard instances for the case that the accuracy parameter $\varepsilon$ is a small universal constant. On the one hand we define $D_{\text{YES}}$ so that all $\mathbf{p}_i$'s are the same except for a "small" number of domain elements i.e., $c \cdot k$ for a small constant $c \in (0, 1)$. On the other hand, for a distribution $\mathbf{p}$ drawn from $D_{\text{NO}}$, $\mathbf{p}_i$ will be randomly 0 or roughly $2/n$, except for at most a constant fraction of the elements. It is not hard to see that, with high probability, a distribution drawn from $D_{\text{YES}}$ (resp. $D_{\text{NO}}$) will be a $k$-histogram (resp. far from being a $k$-histogram).

To ensure that the underlying distributions are indistinguishable using a small sample size, we want to guarantee that, for all small values of $t$, the number of elements with exactly $t$ samples will be roughly the same for $D_{\text{YES}}$ and $D_{\text{NO}}$, as this rules out any test statistic relying on counting the number of $t$-way collisions among the sample. Following [Val11, VV13, JVYHW15, WY16] this is essentially equivalent to showing that distributions drawn from $D_{\text{YES}}$ and $D_{\text{NO}}$ *match their low-degree moments*.

In particular, for a random pair of distributions $\mathbf{p}, \mathbf{p}'$ drawn from $D_{\text{YES}}$ and $D_{\text{NO}}$ respectively, we want that $\sum_i \mathbf{p}_i^t$ and $\sum_i \mathbf{p}'^t_i$ are roughly the same for all small values $t$. We note that the non-exceptional elements of a distribution $\mathbf{p}'$ drawn from $D_{\text{NO}}$ — which have probability mass either 0 or roughly $2/n$ — will have second moment larger than the non-exceptional elements of a distribution $\mathbf{p}$ drawn from $D_{\text{YES}}$ — which have probability mass roughly $1/n$ — by approximately $1/n$. To counteract this discrepancy, the (fewer than $k$) exceptional elements in $D_{\text{YES}}$ must have average mass at least $1/\sqrt{kn}$. Fortunately, using techniques from [VV13, WY$^+$19], we are able to construct distributions that match $t = \widetilde{\Theta}(\log n)$ moments in which no individual bin has mass more than $\widetilde{O}(1/\sqrt{kn})$. Combining this construction with basic information-theoretic arguments gives us an $\widetilde{\Omega}(\sqrt{kn})$ sample complexity lower bound. We note that this lower bound is tight in the sense that with more than $\widetilde{\Omega}(\sqrt{kn})$ samples one can reliably identify the exceptional elements, as they will each have relatively large numbers of samples with high probability, allowing us to distinguish $D_{\text{YES}}$ from $D_{\text{NO}}$ simply based on the sub-distributions over these elements.

Given the aforementioned construction (for constant $\varepsilon$), it is easy to obtain a sample lower bound of $\widetilde{\Omega}(\sqrt{kn}/\varepsilon)$ by mixing our hard instances with the uniform distribution (with mixing weights $\varepsilon$ and $1 - \varepsilon$ respectively). In fact, even if the testing algorithm knows in advance which samples come from the uniform part and which samples come from the original hard instance, distinguishing would require $\widetilde{\Omega}(\sqrt{kn})$ samples from the original hard instance, and therefore $\widetilde{\Omega}(\sqrt{kn}/\varepsilon)$ samples overall. This sample size lower bound turns out to be tight for $\varepsilon$ relatively large, as one can still reliably identify the exceptional bins with only $\widetilde{\Omega}(\sqrt{kn}/\varepsilon)$ samples. However, when $\varepsilon$ becomes sufficiently small, identifying the exceptional bins becomes more challenging. Indeed, if we take $m$ samples, we expect that an exceptional bin has roughly $m\varepsilon/\sqrt{kn}$ more samples than a non-exceptional bin. On the other hand, a non-exceptional bin will have roughly $\text{Poi}(m/n)$ samples with standard deviation $\sqrt{m/n}$. When $m/n \gg m\varepsilon/\sqrt{kn}$ (which happens in the regime $\varepsilon \ll \sqrt{k/n}$), in order for the exceptional bins to be distinguishable, we would need that $m\varepsilon/\sqrt{kn} \gg \sqrt{m/n}$ or $m \gg k/\varepsilon^2$ many samples. Using a careful information-theoretic argument, we formalize this intuition to show that $\widetilde{\Omega}(k/\varepsilon^2)$ is indeed a sample lower bound in this regime.

**Sample-Efficient Tester.** The starting point of our efficient tester is the *Testing-via-Learning* approach of [ADK15]. Very roughly speaking, we first design a learning procedure which outputs a distribution $\hat{\mathbf{p}}$ that would be close to $\mathbf{p}$ in $\chi^2$ divergence, assuming that $\mathbf{p}$ was in fact a $k$-histogram. Then we use a $\chi^2/\ell_1$ tolerant tester, in the spirit of the one introduced in [ADK15], to distinguish between the cases that $\mathbf{p}$ is close to $\hat{\mathbf{p}}$ in $\chi^2$ divergence versus far from $\hat{\mathbf{p}}$ in $\ell_1$-distance. This step is however significantly harder than this simple outline suggests, as it turns out challenging to perform the first step exactly. Instead, we design a specific learning algorithm with an implicit "hybrid" learning guarantee, (see Lemma 5) which in turns requires us to considerably generalize and adapt the "tolerant testing part" to avoid spurious discrepancies (introduced in the imperfect learning stage) which may lead to false negatives.

To implement the first step, we follow the general "learn-and-sieve" idea suggested in [Can16], with important modifications to address the flaw in their approach and its analysis. In particular, suppose that $\mathbf{p}$ is a $k$-histogram. Then, if we *knew* the corresponding $k$ intervals (that make up the partition for the $k$-histogram), it suffices to learn the mass of $\mathbf{p}$ on each interval, and let $\hat{\mathbf{p}}$ be uniform on each interval (with the appropriate total mass). Of course, a key source of difficulty arises from the fact that we do not know the partition a priori. To circumvent this issue, we divide $[n]$ into (roughly) $K = \Theta(k)$ intervals and try to detect if $\mathbf{p}$ is far in $\chi^2$ divergence from being uniform on any of these intervals. If an interval from our partition incurs large $\chi^2$ error (we call such an interval *bad*), we know that $\mathbf{p}$ must not be constant within this interval. Therefore, we proceed to subdivide these bad intervals into roughly equal parts, and recurse on the $\Theta(k)$ intervals in our new partition. Assuming $\mathbf{p}$ is a $k$-histogram, we subdivide at most $k$ intervals in each iteration, since there could be at most $k$ intervals from any interval partition of $[n]$ where $\mathbf{p}$ is not constant. Hence, in each iteration, we decrease the mass of the bad intervals by at least a constant factor. We repeat the process for at most $O(\log(1/\varepsilon))$ many iterations; after this many iterations, the total mass of the bad intervals will become $O(\varepsilon)$, and thus they may be safely ignored.

A significant difference between our method and the approach from [Can16] lies in the method of *sieving*. In [Can16], it was only said that the algorithm would filter out a subset of *breakpoint* intervals based on the $\chi^2$ statistics ([ADK15]) with the goal of reducing discrepancy; this is where

the main gap in their analysis lies, and the particular (flawed) approach they suggested does not seem to be fixable [Can22]. On the contrary, we characterize the exact set of intervals that need to (and can) be removed with the formal definition of *bad* intervals with respect to a given partition $\mathcal{I}$ of $[n]$ (See Definition 2). Based on that, our approach is to search for *any* sub-intervals $J$ (not necessarily an interval in $\mathcal{I}$) on which the $\chi^2$ divergence between $\mathbf{p}$ and $\hat{\mathbf{p}}$, an approximation of $\mathbf{p}$ assuming $\mathbf{p}$ is uniform over intervals within the given partition, is more than $\widetilde{\Omega}(\varepsilon^2/k)$. For an interval $I$ from the partition $\mathcal{I}$, we show the inclusion of such "bad sub-interval" $J \subseteq I$ then certifies the "badness" of $I$ itself. To find such a $J$, we need a technique for accurately approximating $\mathbf{p}(J)$ simultaneously for all intervals $J \subseteq [n]$, in both absolute and relative error; a notion of approximation much stronger that what classical tools from statistical learning theory such as the VC inequality or the Dvoretzky–Kiefer–Wolfowitz (DKW) inequality provide. Notice that, for a *fixed* interval $J \subseteq [n]$, taking the empirical distribution over $b$ samples gives an estimate $\mathbf{q}$ of $\mathbf{p}$ such that $|\mathbf{q}(J) - \mathbf{p}(J)| < \sqrt{\mathbf{p}(J)/b}$ with constant probability. By taking $\Theta(\log(n))$ batches of samples (each containing $b$ i.i.d. samples from $\mathbf{p}$), and computing the median value of all of the $\mathbf{q}(J)$'s, with high probability for each $J$, we then obtain an estimate $\hat{\varphi}(J)$ [4] for which the above condition holds. Using the sub-routine, as long as $b$ is at least $\Omega(k/\varepsilon^2)$, we can ensure that $|\hat{\varphi}(J) - \mathbf{p}(J)|^2/\mathbf{p}(J) \ll \varepsilon^2/k$, and we can then safely use our estimate $\hat{\varphi}(J)$ as a proxy for $\mathbf{p}(J)$ for the detection of those "bad sub-intervals" for which $|\mathbf{p}(J) - \hat{\mathbf{p}}(J)|^2/\hat{\mathbf{p}}(J)$ is large, which in turns certify the bad intervals from a given partition. This suffices *unless* $\mathbf{p}(J)$ is substantially larger than our estimate $\hat{\mathbf{p}}(J)$.

Unfortunately, this bad case where $\mathbf{p}(J) \gg \hat{\mathbf{p}}(J)$ can happen if $\mathbf{p}(J)$ is smaller than $1/b$. In such a case, in a collection of $b$ samples from $\mathbf{p}$, we are likely to see no samples in $J$, and thus our empirical estimate $\hat{\mathbf{p}}(J)$ will be 0. We can fix this issue (i.e., the case where $\hat{\mathbf{p}}(J)$ is actually 0) by mixing both $\mathbf{p}$ and $\hat{\mathbf{p}}$ with the uniform distribution, thus allowing us to assume that $\hat{\mathbf{p}}(J) \geq |J|/2n \geq 1/(2n)$. Yet, this still leaves a potential gap of roughly $n/b$ between the ratio of $\mathbf{p}(J)$ and $\hat{\mathbf{p}}(J)$. Fortunately, if we select $b \gg \sqrt{nk}/\varepsilon$, we will have that $|\hat{\varphi}(J) - \mathbf{p}(J)|^2/\mathbf{p}(J) \ll \varepsilon/\sqrt{nk}$, and even accounting for losing a factor of $b/n$, we will still have that $|\hat{\varphi}(J) - \mathbf{p}(J)|^2/\mathbf{p}(J) \ll \varepsilon^2/k$. This implies that we will successfully detect any bad intervals and achieve our learning guarantees.

## 1.4 Prior Work

Motivated by the question of building provably good succinct representations of a dataset from only a small sub-sample of the data, [ILR12] first introduced histogram testing as a preliminary, ultra-efficient decision subroutine to find the best parameter $k$ for the number of bins. They provided an algorithm for this task which required $\widetilde{O}(\sqrt{kn}/\varepsilon^5)$ samples from the dataset, a sample complexity which beats the naïve approach (reading and processing the whole dataset) for small values of $k$ and relatively large values of the accuracy parameter $\varepsilon$. Subsequent work [CDGR18] reduced the dependence on $\varepsilon$ from quintic to cubic, giving an algorithm with sample complexity $\widetilde{O}(\sqrt{kn}/\varepsilon^3)$. This bound was, however, still quite far from the "trivial" lower bound of $\Omega(\sqrt{n}/\varepsilon^2)$, which follows from a reduction to uniformity testing (i.e., the case $k = 1$) [Pan08].

Prior to the current work, an $\widetilde{O}(\sqrt{n}/\varepsilon^2 + k/\varepsilon^3)$ upper bound and an $\widetilde{\Omega}(\sqrt{n}/\varepsilon^2 + k/\varepsilon)$ lower bound were obtained in [Can16]. While the lower bound is theoretically sound (albeit, as it turns out, suboptimal), as pointed out in [Can22], the upper bound does not hold due to a technical flaw in the analysis, leaving the optimal sample complexity of the problem open for even constant $\varepsilon$. Moreover, the lower bound of [Can16], based on a reduction of histogram testing to the well-studied problem of support size estimation, provably cannot be improved to provide either (i) a quadratic dependence on $\varepsilon$, i.e., $\widetilde{\Omega}(k/\varepsilon^2)$ or (ii) coupling between the two domain parameters $k, n$, i.e., $\widetilde{\Omega}(\sqrt{nk}/\varepsilon)$. Our work remedies all those issues, fully resolving the question of histogram testing, for the whole parameter range, with logarithmic factors.

Finally, we note that a number of works have obtained algorithms and lower bounds for related, yet significantly different, testing problems. Specifically, [DK16] gave a sample-optimal testing algorithm for the special case of our problem where the $k$ intervals are known *a priori*. Moreover, a number of works [DKN15b, DKN15a, DKN17] have obtained identity and equivalence testers *under the assumption* that the input distributions are $k$-histograms.

---

[4]Notice that $\hat{\varphi}$ is neither a distribution nor a measure, but just a map from intervals to positive real values.

**Preliminaries.** We denote by $\text{TV}(\mathbf{p}, \mathbf{q})$ the total variation (TV) distance between probability distributions $\mathbf{p}, \mathbf{q}$ over $[n] := \{1, 2, \ldots, n\}$, defined as $\text{TV}(\mathbf{p}, \mathbf{q}) := \sup_{S \subseteq [n]}(\mathbf{p}(S) - \mathbf{q}(S)) = \frac{1}{2}\sum_{i=1}^{n}|\mathbf{p}(i) - \mathbf{q}(i)|$, where $\mathbf{p}(S) := \sum_{i \in S}\mathbf{p}(i)$. We will make essential use of the $\chi^2$-divergence of $\mathbf{p}$ with respect to $\mathbf{q}$, defined as $d_{\chi^2}(\mathbf{p}\|\mathbf{q}) := \sum_{i=1}^{n}(\mathbf{p}_i - \mathbf{q}_i)^2/\mathbf{q}_i$. We will also require generalizations of these definitions on restrictions of the domain. In particular, given $S \subseteq [n]$, we use the notation $\text{TV}^S(\mathbf{p}, \mathbf{q}) := (1/2)\sum_{i \in S}|\mathbf{p}(i) - \mathbf{q}(i)|$ and $d_{\chi^2}^S(\mathbf{p}\|\mathbf{q}) := \sum_{i \in S}(\mathbf{p}_i - \mathbf{q}_i)^2/\mathbf{q}_i$. We note that for any $S \subseteq [n]$, it holds that $\text{TV}^S(\mathbf{p}, \mathbf{q})^2 \leq \frac{1}{4}d_{\chi^2}^S(\mathbf{p}\|\mathbf{q})$.

The asymptotic notation $\tilde{O}$ (resp. $\tilde{\Omega}$) suppresses logarithmic factors in its argument, i.e., $\tilde{O}(f(n)) = O(f(n)\log^c f(n))$ and $\tilde{\Omega}(f(n)) = \Omega(f(n)/\log^c f(n))$, where $c > 0$ is a universal constant. The notations $\ll$ and $\gg$ intuitively mean "much less than" and "much greater than" respectively. Formally, we write $f(n) \ll g(n)$ to denote that $f(n) < c \cdot g(n)$, for some universal constant $0 < c$.

## 2 An efficient testing algorithm

**A preliminary simplification.** Without loss of generality, we will assume that $\mathbf{p}(i) \geq \frac{1}{2n}$ for every $i \in [n]$. Indeed, this can be ensured by mixing the unknown distribution with the uniform distribution $\mathbf{u}_n$ on $[n]$ beforehand i.e. $\mathbf{p}' := \frac{1}{2}(\mathbf{p} + \mathbf{u}_n)$ (See Fact 3 in Appendix for how to sample from $\mathbf{p}'$ efficiently). It is easy to see that $\mathbf{p}'$ remains a histogram after mixing: $\mathbf{p}' \in \mathcal{H}_k^n$ if $\mathbf{p} \in \mathcal{H}_k^n$, and $\mathbf{p}'$ is at least $(\varepsilon/2)$-far away from every histogram if $\mathbf{p}$ is $\varepsilon$-far from every histogram.

**Testing via Learning.** The main approach is to follow the **Testing-via-Learning** framework proposed in [ADK15]. In particular, suppose we have a learning algorithm capable of constructing $\hat{\mathbf{p}}$ that is close to $\mathbf{p}$ in $\chi^2$ divergence when $\mathbf{p} \in \mathcal{H}_n^k$. Then, (1) if $\mathbf{p} \in \mathcal{H}_k^n$, we will have that $\mathbf{p}$ and $\hat{\mathbf{p}}$ are close *and* (as a consequence of this) that $\hat{\mathbf{p}}$ is close to being a $k$-histogram. Yet, (2) if $\mathbf{p}$ is far from being a $k$-histogram, then by the triangle inequality we must have either that $\hat{\mathbf{p}}$ is far from being a $k$-histogram, or that $\mathbf{p}$ and $\hat{\mathbf{p}}$ are far from each other in $\ell_1$ distance. We can use dynamic programming to check the explicit description is indeed close to a $k$-histogram in $\ell_1$ distance efficiently (See Lemma 4.11 of [CDGR18]). To verify $\mathbf{p}$ and $\hat{\mathbf{p}}$ are close, we will use a result of [ADK15] on tolerant identity testing. In particular, given an explicit description $\hat{\mathbf{p}}$, the tester takes sample from the unknown distribution $\mathbf{p}$ and decides whether $\mathbf{p}$ and $\hat{\mathbf{p}}$ are closed in $\chi^2$ divergence or far in $\ell_1$ distance. We remark that $\hat{\mathbf{p}}$ can be relaxed to be a positive measure.

**Lemma 1** (Adapted from Lemmas 2 and 3 [ADK15]). *Let $\mathbf{p}$ and $\hat{\mathbf{p}}$ be a distribution and a positive measure defined on $[n]$ respectively. Fix $\varepsilon \in (0, 1)$ and let $\mathcal{A} = \{i \in [n] : \hat{\mathbf{p}}(i) \geq \varepsilon/(50n)\}$. There exists a tester **Tolerance-Identity-Test**, which takes $\text{Poi}(m)$ for $m = \Theta\left(\sqrt{n}/\varepsilon^2\right)$ i.i.d. samples from $\mathbf{p}$ and outputs **Accept** if $d_{\chi^2}^{\mathcal{A}}(\mathbf{p}\|\hat{\mathbf{p}}) \leq \varepsilon^2/500$ and **Reject** if $TV^{\mathcal{A}}(\mathbf{p}, \hat{\mathbf{p}}) \geq \varepsilon$ with constant probability.*

**Outline for Learning.** If $\mathbf{p} \in \mathcal{H}_k^n$ and we know the partition of $\mathbf{p}$ in advance, one can learn $\mathbf{p}$ up to $\varepsilon^2$ in $\chi^2$ divergence with $\Theta(k/\varepsilon^2)$ samples (following the analysis of Laplace estimator from [KOPS15]). Without the partition information, we can nonetheless achieve a weaker guarantee. That is, we can output a fully specified measure $\hat{\mathbf{p}}$ on $[n]$, together with a sub-domain $\mathcal{G} \subseteq [n]$, such that $d_{\chi^2}^{\mathcal{G}}(\mathbf{p}\|\hat{\mathbf{p}})$ is small. In particular, we can achieve the guarantee in three steps. (i) Divide the domain $[n]$ into $K \gg k$ many intervals obliviously (Lemma 2). (ii) Output a succinct measure $\hat{\mathbf{p}}$ that is constant on each interval specified by Step (i) (Section 2.1). (iii) Identify the intervals $I$ where $d_{\chi^2}^I(\mathbf{p}\|\hat{\mathbf{p}})$ is large (Section 2.2). Denote $\mathcal{B} = [n]\backslash\mathcal{G}$. The fact that we only have $\mathbf{p}$ and $\hat{\mathbf{p}}$ close in $\chi^2$ divergence on a sub-domain $\mathcal{G}$ is a reasonable compromise as long as $\mathbf{p}(\mathcal{B}), \hat{\mathbf{p}}(\mathcal{B}) \ll \varepsilon$: if $\mathbf{p}$ is $\varepsilon$-far away from $\hat{\mathbf{p}}$ in $\ell_1$ distance on $[n]$, $\mathbf{p}$ is at least $(\varepsilon - \mathbf{p}(\mathcal{B}) - \hat{\mathbf{p}}(\mathcal{B}))$-far away from $\hat{\mathbf{p}}$ on $[n]\backslash\mathcal{B}$. Otherwise, we may take more samples from $\mathbf{p}$ restricted to $\mathcal{B}$ and sub-divide the problematic intervals identified in Step (iii). Repeating the above steps iteratively then brings us to the case $\mathbf{p}(\mathcal{B}) \ll \varepsilon$.

**Equitable Partition.** The first step is to divide the domain into $\Theta(k)$ many intervals over which the masses of $\mathbf{p}$ are approximately equal. As shown in [ADK15], this can be done with $\tilde{\Theta}(k)$ many samples through a routine we denote as **Approx-Divide**. We also need a routine for sub-dividing a set of disjoint intervals into even lighter sub-intervals. Nonetheless, one can reduce the sub-dividing task to domain partitioning by running **Approx-Divide** on the sub-distribution restricted to the set of disjoint intervals. Proofs are provided in Appendix A.1.

**Lemma 2.** *There exists an algorithm **Approx-Sub-Divide** that, given parameters $\delta \in (0, 1]$ and integer $B > 1$, as well as a set of disjoint intervals $\mathcal{I} = \{I_1, I_2, \cdots, I_q\}$, given sample access to $\mathbf{p}$ on $[n]$, outputs a list of partitions $\mathcal{S}_1, \ldots, \mathcal{S}_q$, where $\mathcal{S}_i$ is the partition of the interval $I_i \in \mathcal{I}$, such that the following holds with probability at least $1 - \delta$. (i) The algorithm uses $O\left(B/\mathbf{p}(\mathcal{I}) \cdot \log\left(B/\delta\right)\right)$ samples. (ii) The output contains at most $(8B + q)$ intervals in total. (iii) Every non-singleton interval $S \in \bigcup_{j=1}^{q} \mathcal{S}_j$ satisfies $\mathbf{p}(S) \leq \mathbf{p}(\mathcal{I}) \cdot 16/B$.*

## 2.1 Simultaneously Estimating Mass of Intervals

In this section, we first introduce **Interval-Mass-Estimate**, a sub-routine that can accurately approximate the mass of $\mathbf{p}(J)$ for all intervals $J \subseteq [n]$ simultaneously and then show how we can use it to learn $\mathbf{p}$ (assuming $\mathbf{p} \in \mathcal{H}_n^k$).

**Interval-Mass-Estimate** first divides the number of samples drawn into $\Theta(\log(n/\delta))$ batches. For an interval $I$, we compute the estimate (number of samples falling in $I$ divided by the batch size) for each batch separately and compute the median over the statistics. This is often referred as the "Median Trick" and is crucial in achieving the learning guarantees with high probability. Pseudo-code and analysis are provided in Appendix A.2.

**Lemma 3.** *Let be $\mathbf{p}$ be supported on $[n]$ such that $\mathbf{p}(i) \geq 1/(2n)$. Fix $b \in \mathbb{Z}^+$ and $\delta \in (0, 1]$. The algorithm **Interval-Mass-Estimate** takes $3b \log(n/\delta)$ i.i.d. samples from $\mathbf{p}$ and outputs $\hat{\varphi}$, a map from sub-intervals of $[n]$ to real values, such that, with probability at least $1 - \delta$, for every sub-interval $I \subseteq [n]$ it holds that $\mathbf{p}(I)/\hat{\varphi}(I) \leq \max(2, 8n/b)$, $\hat{\varphi}(I)/\mathbf{p}(I) \leq 3$ and $|\hat{\varphi}(I) - \mathbf{p}(I)| \leq \sqrt{\mathbf{p}(I)/b}$.*

Let $\mathcal{I}$ be a partition of $[n]$. We try to learn $\mathbf{p}$ pretending that $\mathbf{p}$ is constant over each interval within $\mathcal{I}$ with the routine **Empirical-Learning**. In particular, the algorithm uses **Interval-Mass-Estimate** to obtain estimations of the mass of $I \in \mathcal{I}$ and then flatten the mass uniformly among elements $i \in I$. Notice that, due to the application of the median trick, the output is not necessarily a distribution but rather a positive measure[5] $\hat{\mathbf{p}}$ on $[n]$ which is constant over each interval within $\mathcal{I}$.

If $\mathbf{p}$ is indeed a $k$-histogram, errors are only incurred on a special type of intervals (of which there are at most $k$) which we refer to as the *breakpoint intervals*.

**Definition 1** (Breakpoint Intervals)**.** *Given a $k$-histogram $\mathbf{p}$ on $[n]$, we say that $i \in [n]$ is a* breakpoint *with respect to $\mathbf{p}$ if $\mathbf{p}(i) \neq \mathbf{p}(i + 1)$; and that an interval $I \subseteq [n]$ is a* breakpoint interval *(with respect to $\mathbf{p}$) if $I$ contains at least one breakpoint.*

With Definition 1 in mind, we now specify the formal learning guarantees. Pseudo-code and proofs are provided in Appendix A.3.

**Lemma 4.** *Suppose $\mathbf{p} \in \mathcal{H}_k^n$. Let $\mathcal{I}$ a partition of $[n]$ into $K$ intervals. Let $b \in \mathbb{Z}^+$, $\delta \in (0, 1]$ and $T := 3 \log(K/\delta)$. There exists an algorithm **Empirical-Learning**, given $(Tb)$ i.i.d. samples from $\mathbf{p}$, outputs a positive measure $\hat{\mathbf{p}}$, which satisfies the following with probability at least $1 - \delta$. (i) $\hat{\mathbf{p}}$ is constant within each interval in $\mathcal{I}$. (ii) For every sub-intervals $J \subseteq I$ where $I \in \mathcal{I}$ is a non-breakpoint interval with respect to $\mathbf{p}$, we have $\mathbf{p}(J)/\hat{\mathbf{p}}(J) \leq \max(2, 8 \cdot n/b)$ and $|\hat{\mathbf{p}}(J) - \mathbf{p}(J)| \leq \sqrt{\mathbf{p}(J)/b}$.*

By combining the two guarantees in Lemma 4, one can see the $\chi^2$ divergence between $\mathbf{p}$ and $\hat{\mathbf{p}}$, restricted to the non-breakpoint intervals, will be at most $\varepsilon^2$ with high probability if taking $\Theta(KT/\varepsilon^2)$ many samples. However, following a result from [KOPS15, Can16], one only need $\Theta(K/\varepsilon^2)$ samples to learn a $K$-histogram up to $\varepsilon^2$ error in this restricted notion of $\chi^2$ divergence. One may wonder whether this is enough for us, and if the stronger (but less natural) guarantees provided by Lemma 4, which end up increasing the number of samples required, are necessary. As we will see in the next section, we indeed need not only that the $\chi^2$ divergence is small, but also that the ratio $\mathbf{p}(I)/\hat{\mathbf{p}}(I)$ is bounded for all non-breakpoint intervals. In particular, this latter property enables us to compute relatively accurate estimates of the $\chi^2$ divergence restricted to sub-intervals and (consequently) to tell whether $\mathbf{p}$ is constant or from far from being constant on an interval.

## 2.2 Bad Interval Detection

While large contributions to the $\chi^2$ divergence (assuming the learning phase was successful) will only come from breakpoint intervals, not all of them will necessarily contribute significantly to the

---

[5]That is, $\hat{\mathbf{p}}$ might not sum to one, and thus is not itself a probability distribution.

$\chi^2$ divergence. In particular, a breakpoint interval is only considered "bad" and needs to be filtered out if the error incurred is proportional to the number of breakpoints within.

**Definition 2** ($\varepsilon$-Bad-Interval). *Fix a partition $\mathcal{I}$ of $[n]$ containing $K$ intervals. Let $I \in \mathcal{I}$ be a breakpoint interval of $\mathbf{p}$. Furthermore, suppose $I$ contains $j - 1$ breakpoints i.e. $\mathbf{p}$ is $j$-piece-wise uniform in $I$. We say $I \in \mathcal{I}$ is an $\varepsilon$-bad interval with respect to $\hat{\mathbf{p}}$ and $\mathcal{I}$ if $d_{\chi^2}^I \left(\mathbf{p}\|\hat{\mathbf{p}}\right) \geq j \cdot \varepsilon^2/K$.*

The definition suits our purpose for two reasons. (i) The total $\chi^2$ error between $\mathbf{p}$ and $\hat{\mathbf{p}}$ on the set of "good" intervals (complement of the set of "bad" intervals) is small. Indeed, let $\mathcal{G} \in \mathcal{I}$ be a set containing no $\varepsilon$-bad intervals. Since there are at most $K$ intervals contained in $\mathcal{G}$ and $k$ breakpoints contained in the intervals in $\mathcal{G}$, it is easy to see that $d_{\chi^2}^{\mathcal{G}} \left(\mathbf{p}\|\hat{\mathbf{p}}\right) \leq O(\varepsilon^2)$. (ii) One can reliably separate bad intervals from non-breakpoint intervals assuming the learning phase was successful. To see why, note that in that case every non-breakpoint interval $I$ satisfies $d_{\chi^2}^J \left(\mathbf{p}\|\hat{\mathbf{p}}\right) \ll \varepsilon^2/K$ for all $J \subseteq I$ with high probability. On the contrary, for any bad interval $I$, we claim there must be a sub-interval $Q \subseteq I$ where $d_{\chi^2}^Q \left(\mathbf{p}\|\hat{\mathbf{p}}\right) \geq \varepsilon^2/K$ and both $\mathbf{p}$ and $\hat{\mathbf{p}}$ are constant within. In particular, if $I$ is an $\varepsilon$-bad interval that contains $(j - 1)$ breakpoints, we then have a partition $\{Q_1, \cdots, Q_j\}$ of $I$ over which $\mathbf{p}$ is piece-wise constant and at least one of them will have $\chi^2$ error at least $\varepsilon^2/K$.

Our next step is to show how we can leverage the separating condition to design an efficient bad interval detection mechanism. This is where our method *significantly differs* from [Can16]. At a high level, we take another set of independent samples to get an estimate $\hat{\varphi}(Q)$ of $\mathbf{p}(Q)$ for all $Q \subseteq [n]$ simultaneously. Then, we compare $\hat{\varphi}(Q)$ with $\hat{\mathbf{p}}(Q)$ to see whether we have $d_{\chi^2}^Q \left(\mathbf{p}\|\hat{\mathbf{p}}\right) \geq \varepsilon^2/K$, which would in turns imply the interval $I \supseteq Q$ from the given partition is $\varepsilon$-bad. We next provide the pseudo-code for **Learn-And-Sieve**, which finds a positive measure $\hat{\mathbf{p}}$ on $[n]$ and a domain $\mathcal{B}$ such that $d_{\chi^2}^{[n]\backslash\mathcal{B}} \left(\mathbf{p}\|\hat{\mathbf{p}}\right) \leq O(\varepsilon^2)$ provided $\mathbf{p} \in \mathcal{H}_k^n$. Its detailed analysis can be found in Appendix A.4.

---

**Algorithm 1** Learn-And-Sieve

---

**Require:** Sample access to $\mathbf{p}$; a partition $\mathcal{I}$ of $[n]$ containing $K$ intervals; accuracy $\varepsilon$; failure probability $\delta$.
1: Let $m = C \cdot (K/\varepsilon^2 + \sqrt{Kn}/\varepsilon) \cdot \log(n/\delta)$ for a sufficiently large constant $C$.
2: Draw $2m$ i.i.d. samples from $\mathbf{p}$ and split the samples evenly into $\mathcal{S}_1, \mathcal{S}_2$.
3: $\hat{\mathbf{p}} \leftarrow$ **Empirical-Learning** $(\mathcal{S}_1, \mathcal{I}, \delta/4)$, $\hat{\varphi} \leftarrow$ **Interval-Mass-Estimate** $(\mathcal{S}_2, \delta/4)$, $\mathcal{B} \leftarrow \{\}$.
4: **for** all intervals $Q \subseteq I$ for some $I \in \mathcal{I}$ **do**
5:      **if** $\hat{\varphi}(Q)/\hat{\mathbf{p}}(Q) > 6 \cdot \max(1, \varepsilon \cdot \sqrt{n/K})$ **or** $|\hat{\varphi}(Q) - \hat{\mathbf{p}}(Q)| > 0.5\sqrt{\hat{\mathbf{p}}(Q)\varepsilon^2/K}$ **then**
6:          Add $I$ to $\mathcal{B}$.
7: Output Reject if $\mathcal{B}$ contains more than $k$ intervals. Otherwise, **return** $\mathcal{B}, \hat{\mathbf{p}}$.

---

**Lemma 5** (Sieving Lemma). *Given a partition $\mathcal{I}$ containing $K$ intervals, sample access to $\mathbf{p}$ on $[n]$ and $\delta \in (0, 1)$. Then, the output of **Learn-And-Sieve** (Algorithm 1) satisfies the following. (i) Suppose $\mathbf{p} \in \mathcal{H}_k^n$. Then the algorithm returns a positive measure $\hat{\mathbf{p}}$ and $\mathcal{B}$ such that $d_{\chi^2}^{[n]\backslash\mathcal{B}} \left(\mathbf{p}\|\hat{\mathbf{p}}\right) \leq \varepsilon^2$ with probability at least $1 - \delta$. (ii) The output $\mathcal{B}$ contains at most $k$ intervals (if the algorithm does not reject). (iii) At most $O((K/\varepsilon^2 + \sqrt{Kn}/\varepsilon) \cdot \log(n/\delta))$ samples are used.*

*Proof Sketch.* We claim that, if $\mathbf{p} \in \mathcal{H}_k^n$, $\mathcal{B}$ contains all the $\varepsilon$-bad intervals and no non-breakpoint intervals with high probability. Let $I$ be a non-breakpoint interval. For $b = \Theta(m/\log(n/\delta)) = \Theta(K/\varepsilon^2 + \sqrt{Kn}/\varepsilon)$, we have, with high probability, $|\hat{\varphi}(Q) - \mathbf{p}(Q)| \leq \sqrt{\mathbf{p}(Q)/b}$, $|\hat{\mathbf{p}}(Q) - \mathbf{p}(Q)| \leq \sqrt{\mathbf{p}(Q)/b}$ and $\mathbf{p}(Q)/\hat{\mathbf{p}}(Q) \leq \max(2, 8 \cdot n/b)$ which follow from Lemmas 3 and 4. Combining this with triangle inequality and our choice of $b$ implies the second condition of Line 5 will be false. The first condition can be shown to be false by rewriting $\hat{\varphi}(Q)/\hat{\mathbf{p}}(Q)$ as $\hat{\varphi}(Q)/\mathbf{p}(Q) \cdot \mathbf{p}(Q)/\hat{\mathbf{p}}(Q)$, which are themselves bounded, with high probability, by 3 and $\Theta(1) \cdot \max(1, \varepsilon\sqrt{n/K})$ again by Lemmas 3 and 4 and our choice of $b$.

Let $I$ be a breakpoint interval. We then have $|\mathbf{p}(Q) - \hat{\mathbf{p}}(Q)| \geq \sqrt{\hat{\mathbf{p}}(Q) \cdot \varepsilon^2/K}$ for some sub-interval $Q \subset I$. If $\mathbf{p}(Q)$ is light ($\mathbf{p}(Q) \leq 2\varepsilon/\sqrt{Kn}$), we can show $\mathbf{p}(Q)/b \leq 1/4 \cdot \hat{\mathbf{p}}(Q) \cdot \varepsilon^2/K$, making $\hat{\varphi}(Q)$, our estimation for $\mathbf{p}(Q)$, sufficiently accurate such that the second condition of Line 5 will be true. Otherwise, as $b \gg \sqrt{Kn}/\varepsilon$, the estimation $\hat{\varphi}(Q)$ will be within multiplicative factors of $\mathbf{p}(Q)$. If $\hat{\mathbf{p}}(Q)$ is not much lighter than $\mathbf{p}(Q)$, we can again show $\mathbf{p}(Q)/b \leq 1/4 \cdot \hat{\mathbf{p}}(Q) \cdot \varepsilon^2/K$.

Otherwise, the first condition of Line 5 will be true. Conditioned on that $\mathcal{B}$ includes all $\varepsilon$-bad intervals and no non-breakpoint intervals, it is easy to see that $\mathcal{B}$ will contain no more than $k$ intervals and $d_{\chi^2}^{\mathcal{I}\setminus\mathcal{B}}(\mathbf{p}\|\hat{\mathbf{p}}) \le O(\varepsilon^2)$. We note that points (i) and (iii) follow from the definition of the algorithm. $\qquad\square$

**Learn-and-Sieve** (Algorithm 1) outputs a fully specified description $\hat{\mathbf{p}}$ and a sub-domain $\mathcal{G} := [n]\setminus\mathcal{B}$ such that $d_{\chi^2}^{\mathcal{G}}(\mathbf{p}\|\hat{\mathbf{p}})$ is small given $\mathbf{p} \in \mathcal{H}_n^k$. For testing purposes, this is a reasonable divergence from the ideal guarantee that $d_{\chi^2}(\mathbf{p}\|\hat{\mathbf{p}})$ is small *as long as* $\mathbf{p}(\mathcal{B})$ *is also small*. If so, we can set $\hat{\mathbf{p}}(i) = 0$ for $i \in \mathcal{B}$ and invoke **Tolerant-Identity-Test** with $\mathbf{p}$ and $\hat{\mathbf{p}}$. If the test passes, we then know that $\mathrm{TV}^{\mathcal{G}}(\mathbf{p}, \hat{\mathbf{p}}) \le \varepsilon/2$: this together with $\mathbf{p}(\mathcal{B}) \le \varepsilon/2$ then gives $\mathrm{TV}(\mathbf{p}, \hat{\mathbf{p}}) \le \varepsilon$.

Unfortunately, running **Learn-and-Sieve** only once we may have $\mathbf{p}(\mathcal{B}) = \Omega(1)$. To handle this, we will need more fine-grained sieving procedure, which uses **Approx-Sub-Divide** to further partition the bad intervals detected and invokes **Learn-and-Sieve** *iteratively*. In each iteration, the total mass of the bad intervals shrinks by a constant factor, allowing us to reach $\mathbf{p}(\mathcal{B}) \ll \varepsilon$ in at most $O(\log(1/\varepsilon))$ iterations. The pseudo-code (Algorithm 4) and detailed analysis are provided in Appendix A.5.

## 3 Histogram Lower Bound

In this section, we describe the hard instance of histogram testing, which leads to an $\widetilde{\Omega}(\sqrt{kn}/\varepsilon + k/\varepsilon^2)$ lower bound. We will apply the so-called Poissonization trick: we will relax $P$, the unknown object being tested, to be a positive measure with total mass $\Theta(1)$. We denote such a measure as an approximate probability vector and give the corresponding notion of histogram.

**Definition 3** (Approximate Probability Vector). *We define the set of $\nu$-approximate probability vectors (APV) on the domain $[n]$ by $\tilde{\mathcal{P}}^n(\nu) := \{P : P_i \in [0, \infty) \,\forall i \in [n], \big|\|P\|_1 - 1\big| \le \nu\}$. Accordingly, the set of histogram APV is given by $\tilde{\mathcal{H}}_k^n(\nu) := \{P \in \tilde{\mathcal{P}}^n(\nu) : P/\|P\|_1 \in \mathcal{H}_k^n\}$.*

Under the Poisson sampling model, given an unknown $P \in \tilde{\mathcal{P}}^n(\nu)$, the goal it to decide whether $P \in \tilde{\mathcal{H}}_k^n(\nu)$ or $P$ is at least $\varepsilon(1+\nu)$-far[6] from any $P' \in \tilde{\mathcal{H}}_k^n(\nu)$ in $\ell_1$ distance when given the vector $\{M_1, M_2, \cdots M_n\}$ where $M_i \sim \mathrm{Poi}(m \cdot P_i)$. We denote the sample complexity of the problem by $m_{\mathrm{hist}}^{\mathrm{poi}}(n, k, \varepsilon, \nu)$ and provide its formal definition in Appendix B.

To lower bound $m_{\mathrm{hist}}^{\mathrm{poi}}(n, k, \varepsilon, \nu)$, we follow the idea of *moment matching* illustrated in [Val11, VV13, WY16]. In particular, one first constructs two discrete non-negative random variables $U, U'$ whose first few moments are identical. Moreover, $U$ and $U'$ will be designed to have different properties such that one can use i.i.d. copies of $U$ (and $U'$) to generate random measures that are histograms (and far-away from histograms respectively).

Our construction of such a pair of random variables is based on *Chebyshev's polynomials*, a standard tool in approximation theory and the parameter estimation literature. The two variables will be supported on the roots of the polynomial $p(x) = x\left(x - \frac{1}{n}\right)\left(x - \frac{2}{n}\right)T_d\left(1 - \frac{\sqrt{kn}}{C \cdot \log^2 n} \cdot x\right)$, where $T_d(\cdot)$ is the *Chebyshev's polynomial* (of the first kind) and $C$ is a sufficiently large constant. More precisely, $U$ will be supported on roots $r$ where the derivatives $p'(r) < 0$, $U'$ will be on roots where $p'(r) > 0$, and the probabilities will be proportional to $p'(r)$ accordingly. Consequently, $U$ will most likely be $1/n$ (hence useful for histogram construction) and $U'$ will most likely be $0$ or $2/n$, each with non-trivial probabilities (hence appropriate for non-histogram construction). Besides, they will have their maximums bounded by $\widetilde{O}(1/\sqrt{kn})$, which is crucial to achieve the nearly optimal lower bounds. The detailed construction and analysis are provided in Appendix B.1.

**Lemma 6.** *Given positive integers $k, n$ where $k < n$, there exists a pair of non-negative random variable $U, U'$ supported on $[0, 1)$ and absolute constants $c, c' > 0$ satisfying (i) $\Pr\left[U \ne \frac{1}{n}\right] \ll \frac{k}{n}$. (ii) $\Pr\left[U' = 0\right] > 1/3$ and $\Pr\left[U' = \frac{2}{n}\right] > 1/3$. (iii) $U, U' \le c'\log^2 n/\sqrt{kn}$. (iv) $\mathbf{E}[U] = \mathbf{E}[U'] = \frac{1}{n}(1 + O(\sqrt{k/n}))$. (v) $\mathbf{E}[U^t] = \mathbf{E}[U'^t]$ for $1 \le t \le c \cdot \log n$.*

We the proceed to construct two families of Approximate Probability Vectors, one of which belongs to $\tilde{\mathcal{H}}_k^n$ and the other far from it using the random variables stated in Lemma 6. To do so, we

---

[6]The extra $(1+\nu)$ factor is to accommodate the fact that $P$ may not be a distribution i.e. $1 \le \|P\|_1 < (1+\nu)$.

define $H = \left(1/n + \varepsilon U^{(1)}, \cdots, 1/n + \varepsilon U^{(n)}\right)$, $H' = \left(1/n + \varepsilon U'^{(1)}, \cdots, 1/n + \varepsilon U'^{(n)}\right)$ where $U^{(i)}, U'^{(i)}$ are $n$ i.i.d. copies of $U, U'$ in Lemma 6.

We address the two regimes $\sqrt{k/n} \leq \varepsilon \log^2 n$ and $\sqrt{k/n} \geq \varepsilon \log^2 n$ separately. In the former case, the heaviest element among $H$ and $H'$ are roughly $\widetilde{\Theta}(\varepsilon/\sqrt{kn})$. Hence, when the algorithm takes $\widetilde{o}(\sqrt{kn}/\varepsilon)$ samples, it rarely sees any element appearing a large number of times. By the moment-matching property of $U$ and $U'$, the probabilities of seeing some elements appearing for $t$ times for $t \leq \log n$ are almost identical under $H$ and $H'$, therefore making $H$ and $H'$ indistinguishable. In the latter case, we have $\varepsilon U \ll \frac{1}{n}$, implying that no elements in the measures are significantly heavier than the rest. As a result, $H$ and $H'$ are both almost uniform except with a different number of "bumps" (elements that are slightly heavier). Subsequently, the algorithm needs more samples (about $\widetilde{\Omega}(k/\varepsilon^2)$) to tell whether a certain element is heavier than the rest, leading to a phase transition in the sample complexity of the problem. We remark that whether $\widetilde{\Omega}(k/\varepsilon^2)$ or $\widetilde{\Omega}(\sqrt{nk}/\varepsilon)$ dominates depends exactly on the relationship between $\sqrt{k/n}$ and $\varepsilon$ (omitting poly-logarithmic factors). Combining the two regimes then gives us the following lower bound, whose proof is provided in Appendix B.2.

**Proposition 2.** *There exists a constant $\nu \in (0,1)$ such that for any sufficiently large $n$ and $\varepsilon \in (0, 1/10)$, it holds $m_{\mathrm{hist}}^{\mathrm{poi}}(n, k, \varepsilon, \nu) \geq \Omega(\max(\sqrt{kn}/(\varepsilon \log n), k/(\varepsilon^2 \log^3 n)))$.*

Finally, we can easily translate our lower bound result in the Poissonized sampling model to the Multinomial (standard fixed-size) sampling model by a standard reduction. Combining it with the known $\Omega(\sqrt{n}/\varepsilon^2)$ bound (see [Can16, Proposition 4.1]) then concludes our lower bound argument. Formal proofs are given in Appendix B.3.

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
