# A Upper Bound

We provide in this appendix the proofs of some technical lemmas, which were omitted from the main paper due to space constraints.

**Fact 3.** *Let $\mathbf{u}$ be the uniform distribution and $\mathbf{p}$ be an arbitrary distribution among $[n]$. We can sample from $\mathbf{p}' := 1/2(\mathbf{p} + \mathbf{u})$ efficiently with sample access to $\mathbf{p}$.*

## A.1 Proof of Equitable Partition

**Lemma 7** (Claim 1 in [ADK15]). *There exists an algorithm **Approx-Divide** that, given parameters $\delta \in (0, 1]$ and integer $B > 1$, takes $O\big(B \log(B/\delta)\big)$ samples from a distribution $\mathbf{p}$ on $[n]$ and, with probability at least $1 - \delta$, outputs a partition of $[n]$ into $K \leq 8B$ intervals $I_1, I_2, \ldots, I_K$ such that the following holds:*

1. *For each $i \in [n]$ such that $\mathbf{p}(i) > 1/B$, there exists $\ell \in [K]$ such that $I_\ell = \{i\}$. (All "heavy" elements are left as singletons)*

2. *Every other interval is such that $\mathbf{p}(I) \leq 16/B$.*

*Proof.* Let $B > 1$, and consider an arbitrary (unknown) probability distribution $\mathbf{p}$ over $[n]$. Let $\hat{\mathbf{p}}$ be the empirical distribution obtained by taking $m = \left\lceil 18B \ln \frac{12B}{\delta} \right\rceil = O(B \log(B/\delta))$ i.i.d. samples from $\mathbf{p}$, where $C > 0$ is an absolute constant to be chosen later in the proof.

Denote by $i_1 < \cdots < i_T$ the elements $i$ such that $\mathbf{p}(i) > 1/B$; note that $0 \leq T \leq B$ since there can be at most $B$ elements with probability mass greater than $1/B$.

Consider the following (deterministic, and unknown to the algorithm) partition of the domain into consecutive intervals:

- each of $i_1 < \cdots < i_T$ is a singleton interval $J_{\ell_j} = \{i_j\}$;

- setting for convenience $i_0 = -1$ and $i_{T+1} = n + 1$, define the remaining intervals greedily as follows. For each $1 \leq j \leq T$, to partition $\{i_j + 1, i_j + 2, \ldots, i_{j+1} - 1\}$, do

   1. set $J \leftarrow \emptyset, t \leftarrow i_j + 1$
   2. while $t < i_{j+1}$
      - if $\mathbf{p}(J \cup \{t\}) < 3/(2B)$, set $J \leftarrow J \cup \{t\}$
      - else add $J$ as an interval to the partition, then start a new interval: $J \leftarrow \emptyset$
      - consider the next element: $t \leftarrow t + 1$

That is, every remaining interval ("in-between" two heavy elements $i_j, i_{j+1}$) is greedily divided from left to right into maximal subintervals of probability mass at most $3/(2B)$. Note that since every element $i \in \{i_j + 1, \ldots, i_{j+1} - 1\}$ has mass at most $1/B$, this is indeed possible and leads to a partition of $\{i_j + 1, \ldots, i_{j+1} - 1\}$ in intervals, where all but at most one (the rightmost one) has probability mass $\mathbf{p}(J) \in [1/(2B), 3/(2B)]$.

Overall, this process (which the algorithm cannot directly run, not knowing $\mathbf{p}$) guarantees the existence of a partition $J_1, \ldots J_S$ of the domain in at most

$$S \leq T + T + 2B \leq 4B$$

consecutive, disjoint intervals (the $T$ singleton intervals, the at most $T$ "rightmost, in-between" intervals which may have probability mass at most $1/(2B)$, and the remaining "in-between" intervals which all have mass at least $1/(2B)$, and of which there can thus be at most $2B$ in total).

Now, consider the following partition $\hat{J}_1, \ldots \hat{J}_K$ defined by the algorithm from $\hat{\mathbf{p}}$:

- every element $i \in [n]$ such that $\hat{\mathbf{p}}(i) > 1/(2B)$ is added as a singleton interval; let their number be $T'$;

- then the algorithm proceeds as in the above process (which defined $J_1, \ldots J_S$), but using $\hat{\mathbf{p}}$ instead of $\mathbf{p}$.

The same analysis (but with $T' \leq 2B$ instead of $T \leq B$, since the threshold for "singletons" is now $1/(2B)$ shows that this will result in a partition of the domain into $K$ consecutive disjoint intervals $\hat{J}_1, \ldots \hat{J}_K$, with

$$K \leq 8B.$$

It remains to argue about the properties of $\hat{J}_1, \ldots \hat{J}_K$, by using those of (the unknown) $J_1, \ldots J_S$. First, we have that, for every $1 \leq j \leq T$, by a Chernoff bound,

$$\Pr\left[\hat{\mathbf{p}}(i_j) \leq \frac{1}{2B}\right] \leq e^{-(1/2)^2 \frac{m}{2B}} \leq \frac{\delta}{3B}$$

from our setting our $m$. Now, for any interval $J_j$ (where $1 \leq j \leq S$) such that $1/(2B) \leq \mathbf{p}(J_j) \leq 3/(2B)$ (there are at most $2B$ of them), we also have, all by a Chernoff bound,

$$\Pr\left[\hat{\mathbf{p}}(i_j) \notin \left[\frac{1}{4B}, \frac{2}{B}\right]\right] \leq 2e^{-\frac{m}{18B}} \leq \frac{\delta}{6B}$$

while for any $J_j$ such that $\mathbf{p}(J_j) < 1/(2B)$ (there are at most $T$ of them)

$$\Pr[\hat{\mathbf{p}}(i_j) \geq 1/B] \leq e^{-\frac{m}{6B}} \leq \frac{\delta}{3B}.$$

This means, by a union bound, that with probability at least $1 - \left(T \cdot \frac{\delta}{3B} + 2B \cdot \frac{\delta}{6B} + T \cdot \frac{\delta}{3B}\right) \geq 1 - \delta$:

- all of the $(1/B)$-heavy elements for $\mathbf{p}$ will also be $(1/(2B))$-heavy elements for $\hat{\mathbf{p}}$, and thus constitute their own singleton interval, as desired;

- all of the intervals $J_j$ such that $1/(2B) \leq \mathbf{p}(J_j) \leq 3/(2B)$ (call those "heavy") satisfy $1/(4B) \leq \hat{p}(J_j) \leq 2/B$;

- all of the intervals $J_j$ such that $\mathbf{p}(J_j) < 1/(2B)$ (call those "light") satisfy $\hat{p}(J_j) \leq 2/B$.

Conditioning on this happening, we can analyze the properties of $\hat{J}_1, \ldots \hat{J}_K$. Specifically, fix any $\hat{J}_j$ which is not a singleton of the form $\{i_\ell\}$ for one of the $T$ $(1/B)$-heavy elements for $\mathbf{p}$. We want to argue that $\mathbf{p}(\hat{J}_j) \leq 16/B$.

- If $\hat{J}_j$ intersects with at most 8 $J_{j'}$'s (which must then all be consecutive, and at least 7 are "heavy", and the rightmost one is either heavy or light), we are done, since then $\mathbf{p}(\hat{J}_j) \leq 8 \cdot 2/B = 16/B$.

- Moreover, $\hat{J}_j$ cannot intersect with more than 8 $J_{j'}$'s, as otherwise those are consecutive, and thus $\hat{J}_j$ must contain at least 6 "heavy" $J_{j'}$'s. But in that case $\hat{\mathbf{p}}(\hat{J}_j) \geq 6 \cdot 1/(4B) = 3/(2B)$, while by our greedy construction we ensured that $\hat{\mathbf{p}}(\hat{J}_j) < 3/(2B)$.

This concludes the proof. $\qquad\square$

*Proof of Lemma 2.* Reduction from Lemma 7. $\qquad\square$

## A.2 Proof of Interval Mass Estimate

---

**Algorithm 2** Interval-Mass-Estimate

---

**Require:** $m$ i.i.d. samples from distribution $\mathbf{p}$ on $[n]$; failure probability $\delta$.
1: Let $T := 3\log(n/\delta)$.
2: Split the samples into $T$ batches $S^{(1)}, S^{(2)}, \cdots S^{(T)}$, each of $m/T$ samples.
3: Initialize a Map $\hat{\varphi} : \mathcal{I} \mapsto \mathbb{R}$ which maps sub-intervals of $[n]$ to real values.
4: **for** all intervals $I \subseteq [n]$ **do**
5:     Let $X_j^{(i)}$ be the number of samples falling in interval $I_j$ from samples $S^{(i)}$.
6:     Compute the median $X_j = \text{Median}_{1 \leq i \leq T}\left(X_j^{(i)}\right)$. Accordingly, set

$$\hat{\varphi}(I) = \max\left(\frac{X_j}{m}, \frac{|I|}{2n}\right).$$

7: Output the map $\hat{\varphi}$.

---

**Claim 4.** *Let $I$ be an interval (or subset) over $[n]$ and $\mathbf{p}$ a distribution over $[n]$. For $\delta \in (0,1)$, assume one takes $T := 3\log(1/\delta)$ batches of i.i.d. samples from $\mathbf{p}$ where each batch is of size $b \in \mathbb{Z}^+$. Denote by $X_I^{(i)}$ the number of samples falling in $I$ from the $i$-th batch, for $i \leq [T]$. Then, the median $X_I := \text{Median}_i(X_I^{(i)})$ satisfies*

$$\left|\frac{X_I}{b} - \mathbf{p}(I)\right| \leq 3 \cdot \sqrt{\frac{\mathbf{p}(I)}{b}} \quad \text{and} \quad \frac{X_I}{b} \leq 3 \cdot \mathbf{p}(I),$$

*with probability at least $1 - \delta$.*

*Proof of Lemma 3.* Let $I \subseteq [n]$ be an arbitrary interval and $X_I$ be the median of the numbers of samples falling in from each batch. By Claim 4, we have that

$$\left|\frac{X_I}{m} - \mathbf{p}(I)\right| \leq \sqrt{\mathbf{p}(I)/b}, \tag{1}$$

$$\frac{X_I}{m} \leq 3\mathbf{p}(I). \tag{2}$$

with probability at least $1 - \delta/(2n)$. Since $\mathbf{p}(i) > 1/(2n)$ for any $i \in [n]$, the max operation decreases the distance between $\mathbf{p}$ and $\hat{\mathbf{p}}$ pointwise. Hence, with probability at least $1 - \delta/2K$, we have

$$|\hat{\varphi}(I) - \mathbf{p}(I)| \leq \sqrt{\mathbf{p}(I)/b}. \tag{3}$$

Since there are at most $n^2$ sub-intervals $I \subseteq [n]$ overall, by a union bound over all sub-intervals, we have that Equation (3) holds simultaneously with probability at least $1 - \delta/2$ for every sub-interval $I$.

To show that the ratio $\mathbf{p}(I)/\hat{\varphi}(I)$ is bounded, we first consider the case $\mathbf{p}(I) \leq 4/b$. Since $\hat{\varphi}(I) > 1/(2n)$, it is easy to see that $\mathbf{p}(I)/\hat{\varphi}(I) \leq 8 \cdot b/n$. Otherwise, we have $\mathbf{p}(I) > 4/b$. Following Equation (3). it holds

$$|\mathbf{p}(I) - \hat{\varphi}(I)| \leq \sqrt{\mathbf{p}(I)/b} \leq \frac{1}{2}\mathbf{p}(I),$$

This then implies that

$$\frac{\mathbf{p}(I)}{\hat{\varphi}(I)} \leq 2.$$

Lastly, since $\mathbf{p}(I) \geq |I|/(2n)$, together with Equation (2), we have

$$\frac{\hat{\varphi}(I)}{\mathbf{p}(I)} = \frac{\max\left(|I|/(2n), X_I/m\right)}{\mathbf{p}(I)} \leq 3.$$

$\square$

## A.3 Proof of Empirical Learning

---

**Algorithm 3** Empirical-Learning

---

**Require:** $m$ i.i.d. samples from distribution $\mathbf{p}$ on [n]; partition $\mathcal{I} = \{I_1, I_2, \cdots, I_K\}$ of the domain $[n]$; error rate $\delta$
1: Denote the $m$ i.i.d samples as $\mathcal{S}$.
2: Construct the Interval Estimator $\mathcal{P} \leftarrow$ **Interval-Mass-Estimate**$(\mathcal{S}, \delta)$.
3: **for** $j = 1 \cdots K$ **do**
4:      Compute the estimate $\hat{\mathbf{p}}(i) = \frac{\mathcal{P}(I_j)}{|I_j|}$     $1 \leq j \leq K, i \in I_j$.
5: Output the measure $\hat{\mathbf{p}}$.

---

## A.4 Detailed Analysis of Learn-And-Sieve

*Proof of Lemma 4.* Let $I$ be a non-breakpoint interval. It is easy to see that $\hat{\mathbf{p}}(I) = \hat{\varphi}(I)$. By Lemma 3, it holds that

$$\frac{\mathbf{p}(I)}{\hat{\mathbf{p}}(I)} \leq \max(2, 8 \cdot n/b),$$

$$|\hat{\mathbf{p}}(I) - \mathbf{p}(I)| \leq \sqrt{\mathbf{p}(I)/b}.$$

with probability at least $1 - \delta$. Condition on that, we easily have, for any sub-interval $J \subseteq I$,

$$\frac{\mathbf{p}(J)}{\hat{\mathbf{p}}(J)} = \frac{\mathbf{p}(I)}{\hat{\mathbf{p}}(I)} \leq \max(2, 8 \cdot n/b)$$

since both $\mathbf{p}$ and $\hat{\mathbf{p}}$ are uniform within $I$. Furthermore, we also have

$$|\mathbf{p}(J) - \hat{\mathbf{p}}(J)| = |\mathbf{p}(I) - \hat{\mathbf{p}}(I)| \cdot \frac{|J|}{|I|} \leq \sqrt{\frac{\mathbf{p}(I)}{m}} \cdot \frac{|J|}{|I|} \leq \sqrt{\frac{\mathbf{p}(J)}{m}},$$

where the first equality follows from the fact that $\mathbf{p}$ and $\hat{\mathbf{p}}$ are both uniform within $I$, the second inequality follows from our conditioning, the third inequality follows from $|J| \leq |I|$. $\qquad\square$

*Proof of Lemma 5.* The first point is easy to see since otherwise Line 7 would reject. We claim that the following holds with probability at least $1 - \delta$. If $\mathbf{p} \in \mathcal{H}_k$, $\mathcal{B}$ contains all the $\varepsilon$-bad intervals and no non-breakpoint intervals. Condition on that, it is easy to see $\mathcal{B}$ would contain no more than $k$ intervals since there are at most $k$ break-point intervals. Besides, for any $I \in \mathcal{G} := \mathcal{I} \backslash \mathcal{B}$, we have $d_{\chi^2}^I (\mathbf{p} \| \hat{\mathbf{p}}) \leq \varepsilon^2/K$. Overall, it then holds $d_{\chi^2}^{\mathcal{I} \backslash \mathcal{B}} (\mathbf{p} \| \hat{\mathbf{p}}) \leq \varepsilon^2$. This concludes the proof of the second point.

It remains to show why $\mathcal{B}$ separates the bad intervals from the rest. We will separately establish that with high probability $\mathcal{B}$ will not include any non-breakpoint intervals and that it will include all bad intervals separately, and conclude by a union bound. The following statement, which follows from Lemma 3, will be useful in both aspects of the analysis. In particular, it holds

$$|\hat{\varphi}(Q) - \mathbf{p}(Q)| \leq \sqrt{\mathbf{p}(Q)/b} \tag{4}$$
$$\hat{\varphi}(Q) \leq 3 \cdot \mathbf{p}(Q) \tag{5}$$

for all sub-intervals $Q$ with probability at least $1 - \delta/2$. We will condition on that in the analysis.

**Exclusion of non-breakpoint intervals** By Lemma 4, if $I$ is not a break-point interval, then for every sub-interval $Q$, with probability at least $1 - \delta/3$, we have

$$|\hat{\mathbf{p}}(Q) - \mathbf{p}(Q)| \leq \sqrt{\mathbf{p}(Q)/b} \tag{6}$$
$$\frac{\mathbf{p}(Q)}{\hat{\mathbf{p}}(Q)} \leq \max(2, 8 \cdot n/b), \tag{7}$$

As a result, we have

$$(\hat{\varphi}(Q) - \hat{\mathbf{p}}(Q))^2 \leq 4 \cdot \mathbf{p}(Q)/b \leq O(1/b + n/b^2) \cdot \hat{\mathbf{p}}(Q) \leq \frac{1}{4}\hat{\mathbf{p}}(Q) \cdot \varepsilon^2/K\,,$$

where the first inequality follows Equations (6), (4) and the triangle inequality, the second inequality follows from the bounded ratio between $\hat{\mathbf{p}}(Q)$ and $\mathbf{p}(Q)$ (Equation (7)) and the last inequality follows from $b \gg \sqrt{nK}/\varepsilon + K/\varepsilon^2$. This asserts that second condition of Line 5 will be false.

Next, we shift our focus to the first condition of Line 5. Together with Equation (7), which is the condition in Lemma 4, and Equation (5), the guarantee obtained from Claim 3, we have

$$\frac{\hat{\varphi}(Q)}{\hat{\mathbf{p}}(Q)} = \frac{\hat{\varphi}(Q)}{\mathbf{p}(Q)} \cdot \frac{\mathbf{p}(Q)}{\hat{\mathbf{p}}(Q)} \leq \max\left(6\,, 24 \cdot n/b\right). \leq \max\left(6\,, \varepsilon\sqrt{n/K}\right).$$

where the last inequality follows from $b \gg \sqrt{nK}/\varepsilon$. This asserts that Line 5 won't pass. Hence, with probability at least $1 - \delta$, $\mathcal{B}$ won't include any non-breakpoint intervals.

**Inclusion of $\varepsilon$-bad intervals** Let $I$ be an $\varepsilon$-bad interval. Then, there exists a sub-interval $Q$ such that $\mathbf{p}$ is constant within $Q$ and $\frac{(\mathbf{p}(Q)-\hat{\mathbf{p}}(Q))^2}{\hat{\mathbf{p}}(Q)} \geq \varepsilon^2/K$, which implies that

$$|\mathbf{p}(Q) - \hat{\mathbf{p}}(Q)| \geq \sqrt{\hat{\mathbf{p}}(Q) \cdot \frac{\varepsilon^2}{K}}. \tag{8}$$

We then proceed to analyze the cases $\mathbf{p}(Q) \leq 2 \cdot \frac{\varepsilon}{\sqrt{Kn}}$ and $\mathbf{p}(Q) > 2 \cdot \frac{\varepsilon}{\sqrt{Kn}}$ separately.

In the former case, we have

$$\frac{\mathbf{p}(Q)}{b} = \frac{\hat{\mathbf{p}}(Q)}{b} \cdot \frac{\mathbf{p}(Q)}{\hat{\mathbf{p}}(Q)} \leq \hat{\mathbf{p}}(Q) \cdot O\left(\varepsilon \cdot \sqrt{\frac{n}{K}}\right) \cdot \frac{1}{b} \leq \frac{1}{4}\hat{\mathbf{p}}(Q) \cdot \frac{\varepsilon^2}{K}\,,$$

where the first inequality follows from that $\mathbf{p}(Q) \leq 2\varepsilon/\sqrt{nK}$ and $\hat{\mathbf{p}}(Q) \geq \frac{1}{2n}$, and the second inequality follows from $b \gg \sqrt{nK}/\varepsilon$. This together with Equations (4), (8) and triangle inequality gives

$$|\hat{\mathbf{p}}(Q) - \hat{\varphi}(Q)| \geq \frac{1}{2}\sqrt{\hat{\mathbf{p}}(Q) \cdot \frac{\varepsilon^2}{K}}\,,$$

which asserts that the second if condition of Line 5 is true.

In the latter case, since $\mathbf{p}(Q) > 2\frac{\varepsilon}{\sqrt{nK}} \gg \frac{1}{b}$, it holds $\frac{\mathbf{p}(Q)}{b} \ll \mathbf{p}(Q)^2$. From Equation (4) we have

$$\hat{\varphi}(Q) \geq \mathbf{p}(Q) - \sqrt{\mathbf{p}(Q)/b} \geq \frac{1}{2}\mathbf{p}(Q). \tag{9}$$

If $\hat{\varphi}(Q)/\hat{\mathbf{p}}(Q) \geq \max(6, \varepsilon\sqrt{n/K})$, the first condition of Line 5 is true, implying $I$ will be included in $\mathcal{B}$. Otherwise, we could argue that

$$\frac{\mathbf{p}(Q)}{b} = \frac{\hat{\mathbf{p}}(Q)}{b} \cdot \frac{\mathbf{p}(Q)}{\hat{\varphi}(Q)} \cdot \frac{\hat{\varphi}(Q)}{\hat{\mathbf{p}}(Q)} \leq \frac{\hat{\mathbf{p}}(Q)}{b} \cdot 2 \cdot \max(6, \varepsilon\sqrt{n/K}) \leq \frac{1}{4}\hat{\mathbf{p}}(Q)\varepsilon^2/K\,,$$

where the first inequality follows from Equation (9) and the fact that the first condition of Line 5 is false, and the second inequality follows from $b \gg \sqrt{nK}/\varepsilon + K/\varepsilon^2$. In other words, we have

$$\sqrt{\frac{\mathbf{p}(Q)}{b}} \leq \frac{1}{2}\sqrt{\hat{\mathbf{p}}(Q) \cdot \frac{\varepsilon^2}{K}}.$$

This together with Equations (4), (8) and triangle inequality allows us to conclude that

$$|\hat{\mathbf{p}}(Q) - \hat{\varphi}(Q)| \geq \frac{1}{2}\sqrt{\hat{\mathbf{p}}(Q) \cdot \frac{\varepsilon^2}{K}}\,,$$

which asserts that the second condition of Line 5 is true. $\qquad\square$

### A.5 Main Testing Algorithm

---

**Algorithm 4** Divide-And-Learn-And-Sieve

---

**Require:** Sample access to the distribution $\mathbf{p}$; domain size $[n]$; accuracy $\varepsilon$.
1: Set $\mathcal{I}^{(0)} = \mathcal{B}^{(0)} = \{[n]\}$.
2: Set $T \leftarrow 3\log(1/\varepsilon)$, $\delta \leftarrow \frac{1}{100T}$, $t \leftarrow 0, r = 1$.
3: **while** $r > \varepsilon/8$ **do**
4:      $\mathcal{S}^{(t+1)} \leftarrow$ **Approx-Sub-Divide** $\left(32k, \mathcal{B}^{(t)}, \delta\right)$.
5:      Set $\mathcal{I}^{(t+1)} = (\mathcal{I}^{(t)} \setminus \mathcal{B}^{(t)}) \cup \mathcal{S}^{(t+1)}$.          ▷ Note that $\mathcal{I}^{(t+1)}$ is still a partition of $[n]$.
6:      $\mathcal{Q}^{(t+1)}, \hat{\mathbf{p}}^{(t+1)} \leftarrow$ **Learn-And-Sieve** $\left(\mathcal{I}^{(t+1)}, \frac{\varepsilon}{4\sqrt{T}}, \delta\right)$.
7:      Reject if **Learn-And-Sieve** outputs Reject.
8:      Set $\mathcal{B}^{(t+1)} = \mathcal{S}^{(t+1)} \cap \mathcal{Q}^{(t+1)}$.
9:      Take $\ell = \Theta\left(\log(1/\delta)/\varepsilon\right)$ i.i.d. samples from $\mathbf{p}$.
10:     Denote $X$ as the number of samples falling in $\mathcal{B}^{(t+1)}$. Set $r = X/\ell$.
11:     $t \leftarrow t + 1$
12: Denote $\mathcal{G}^{(j)} = \mathcal{S}^{(j)} \setminus \mathcal{B}^{(j)}$ for all $j \geq 1$.     ▷ Note that the union of $\mathcal{G}^{(1)}, \cdots \mathcal{G}^{(t)}, \mathcal{B}^{(t)}$ forms a partition of the domain $[n]$.
13: We will consider the measure $\bar{\mathbf{p}}$ such that on intervals $I \in \mathcal{G}^{(j)}$, $\bar{\mathbf{p}}(i) = \hat{\mathbf{p}}^{(j)}(i)$; and on intervals $I \in \mathcal{B}^{(t)}$, $\bar{\mathbf{p}}(i) = 0$.
14: Use Dynamic Programming to check whether there is a $k$-histogram that is $\varepsilon/2$-close to $\bar{\mathbf{p}}$ in $\ell_1$ distance. Otherwise Reject. ▷ Can be done in time $\mathrm{poly}(k, 1/\varepsilon, n)$ as shown in Lemma 4.11 of [CDGR18].
15: Output Accept if **Tolerant-Identity-Test** $(\mathbf{p}, \hat{\mathbf{p}}, \varepsilon)$ outputs Accept. Otherwise Reject.

---

*Proof of Upper Bound in Theorem 1.* We first argue that the algorithm terminates in $T = O(\log(1/\varepsilon))$ rounds with high probability. By Lemma 7, we have $\mathbf{p}(I) \leq \frac{16}{32k} \cdot \mathbf{p}(\mathcal{B}^{(t)}) = \frac{1}{2k} \cdot \mathbf{p}(\mathcal{B}^{(t)})$ for every non-singleton interval $I \in \mathcal{S}^{(t+1)}$ with probability at least $1 - \delta$. By Lemma 5, the subroutine **Learn-And-Sieve** selects (removes) at most $k$ intervals if it does not output reject. Notice that $\mathcal{B}^{(t+1)}$ will not include any singleton as singletons cannot be break-point intervals. Then, it holds that $\mathbf{p}(\mathcal{B}^{(t+1)}) \leq \mathbf{p}(\mathcal{Q}^{(t+1)}) \leq \frac{1}{2}\mathbf{p}(\mathcal{B}^{(t)})$. Hence, the mass of $\mathcal{B}^{(t)}$ will drop below $\varepsilon/100$ after at most $T := 3 \cdot \log(1/\varepsilon)$ iterations with probability at least

$$(1 - \delta)^T = (1 - \frac{1}{100T})^T \geq \frac{9}{10}, \tag{10}$$

where the second inequality holds when $T$ is sufficiently large. On the other hand, in the end of iteration, we always $r \leq 10 \cdot \mathbf{p}(\mathcal{B}^{(t)}) \leq \varepsilon/10$ with probability at least $\frac{9}{10}$ by Markov's Inequality. Combining the two facts then gives the algorithm exits the while loop in at most $T := 3 \cdot \log(1/\varepsilon)$ iterations with probability at least $0.9 \cdot 0.9 > 8/10$.

Furthermore, we claim, with probability at least $9/10$, it holds $\mathbf{p}(\mathcal{B}^{(t)}) \leq \varepsilon/4$ when the algorithm exits the while loop. Suppose at the $j$-th iteration, we have $\mathbf{p}(\mathcal{B}^{(j)}) > \varepsilon/4$. Then, by the multiplicative Chernoff Bound, we have

$$\Pr\left[r \leq \varepsilon/8\right] \leq \Pr\left[r \leq 1/2 \cdot \mathbf{p}(\mathcal{B}^{(j)})\right] \leq 1/\delta.$$

Hence, following the same calculation as Equation (10), our claim holds with probability at least $9/10$. Conditioning on (i) the algorithm terminates in $T$ iterations and (ii)

$$\mathbf{p}(\mathcal{B}^{(t)}) \leq \varepsilon/4 \tag{11}$$

when the algorithm exits the loop, we proceed to argue it outputs the correct testing result with probability at least $2/3$.

**Completeness** Suppose we have $\mathbf{p} \in \mathcal{H}_k^n$. At the $t$-th iteration, we claim that, with probability at least $1 - \delta$, it holds

$$d_{\chi^2}^{\mathcal{G}^{(t)}}\left(\mathbf{p}\|\hat{\mathbf{p}}^{(t)}\right) \leq \frac{\varepsilon^2}{16T}, \tag{12}$$

where $\mathcal{G}^{(t)} = \mathcal{S}^{(t)} \backslash \mathcal{B}^{(t)}$ as defined in 12 and $\hat{\mathbf{p}}^{(t)}$ is the learned distribution in the $t$-th iteration. By Lemma 5, it holds

$$d_{\chi^2}^{\mathcal{I}^{(t)} \backslash \mathcal{Q}^{(t)}} \left( \mathbf{p} \| \hat{\mathbf{p}}^{(t)} \right) \leq \frac{\varepsilon^2}{16T}.$$

Since $\mathcal{G}^{(t)} = \mathcal{S}^{(t)} \backslash \mathcal{B}^{(t)}$ is a subset of $\mathcal{I}^{(t)} \backslash \mathcal{Q}^{(t)}$, the claim in Eq. (12) follows. Recall that we condition on the algorithm runs for at most $T$ iterations. Combining this with Eq. (12), if we denote $\mathcal{G} = \bigcup \mathcal{G}^t$ for $1 \leq t \leq T$, it holds

$$d_{\chi^2}^{\mathcal{G}} \left( \mathbf{p} \| \bar{\mathbf{p}} \right) \leq \sum_{t=1}^{T} d_{\chi^2}^{\mathcal{G}^{(t)}} \left( \mathbf{p} \| \hat{\mathbf{p}}^{(t)} \right) \leq \varepsilon^2 / 16 \tag{13}$$

with probability at least $(1 - \delta)^T \geq \frac{9}{10}$. Notice that $\mathcal{G}$ is precisely the sub-domain $\mathcal{A} = \{i \in [n] : \bar{\mathbf{p}}(i) \geq \varepsilon/(50n)\}$ that will be used to compute the $\chi^2$ statistic since $\bar{\mathbf{p}}(i) \geq \frac{1}{2n}$ for $i \in \mathcal{G}$ and $\bar{\mathbf{p}}(i) = 0$ for $i \notin \mathcal{G}$. Conditioning on Equation (13), by Proposition 1, line 15 will output accept with probability at least $9/10$ by Chebyshev's Inequality. Then, Equation (13) together with the conditioning $\mathbf{p}(\mathcal{B}^{(t)}) \leq \varepsilon/4$ also implies $\mathrm{TV}(\mathbf{p}, \bar{\mathbf{p}}) \leq \varepsilon/2$ and $\mathbf{p} \in \mathcal{H}_k^n$, line 14 will also pass. Overall, the algorithm accepts with probability at least $2/3$.

**Soundness** Suppose now that $\mathrm{TV}(\mathbf{p}, H) > \varepsilon$ for every $H \in \mathcal{H}_k^n$. For the sake of contradiction, assume that line 14 and line 15 both pass. By line 15 and the contra-positive of Proposition 1, we have $\mathrm{TV}^{\mathcal{G}}(\mathbf{p}, \bar{\mathbf{p}}) \leq \varepsilon/4$ with probability at least $\frac{9}{10}$.

By definition we have $\bar{\mathbf{p}}(\mathcal{B}^{(t)}) = 0$. Since $\mathbf{p}(\mathcal{B}^{(t)}) \leq \varepsilon/4$ by our conditioning, it then holds that $\mathrm{TV}(\mathbf{p}, \bar{\mathbf{p}}) \leq \varepsilon/2$. Then, by line 14, there exists a $k$-histogram $\mathbf{h}$ satisfying $\mathrm{TV}(\mathbf{h}, \bar{\mathbf{p}}) \leq \varepsilon/2$. By the triangle inequality, we have $\mathrm{TV}(\mathbf{h}, \mathbf{p}) \leq \varepsilon$. This contradicts the assumption that $\mathbf{p}$ is more than $\varepsilon$-far from any $k$-histogram. Hence, at least one of the two lines will output reject with probability at least $2/3$.

Finally, the samples are consumed by five different types of routines – dividing (Line 4), learning, sieving (Line 6), testing (Line 15) and mass estimation (Line 10). By Lemma 2, in one iteration, the dividing phase consumes $O\left(k \log k / \mathbf{p}\left(\mathcal{B}^{(t)}\right)\right)$ samples where $\mathbf{p}(\mathcal{B}^{(t)})$ is the mass of the to-be-divided intervals at the $t$-th iteration. Since $\mathbf{p}(\mathcal{B}^{(t)})$ shrinks exponentially in every iteration, the samples consumed are dominated by the last iteration. Hence, at most $O\left(k \log k / \varepsilon\right)$ samples are consumed by the dividing phase in total.

At the $t$-th iteration, the partition size $K := \left|\mathcal{I}^{(t+1)}\right|$ is upper bounded by $O(T \cdot k)$. Hence, by Lemma 5, the **Learn-And-Sieve** procedure consumes in total

$$O\left(\left(\frac{Tk}{\varepsilon^2} + \frac{\sqrt{Tkn}}{\varepsilon}\right) \cdot \log\left(n \cdot \log(1/\varepsilon)\right) \cdot T\right) \tag{14}$$

samples. The process of testing the mass of $\mathcal{B}^{(t)}$ consumes $\Theta(T \cdot \log(1/T)/\varepsilon)$ samples in total.

After the algorithm exits the for loop, the chi-square tester consumes $\Theta(\sqrt{n}/\varepsilon^2)$ samples. Thus, in total, the algorithm consumes

$$O\left(\frac{\sqrt{n}}{\varepsilon^2} + \left(\frac{k}{\varepsilon^2} \cdot \log^2(1/\varepsilon) + \frac{\sqrt{kn}}{\varepsilon} \cdot \log^{3/2}(1/\varepsilon)\right) \cdot \log\left(n \cdot \log(1/\varepsilon)\right)\right) \tag{15}$$

if we sum up the samples consumed by all different routines and substitute $T = 3\log(1/\varepsilon)$. $\square$

# B Lower Bound

**Definition 4** (Histogram Testing under Poisson Model). *For $0 < \nu < 1$, $0 < \varepsilon < 1$, define the sample complexity of histogram testing (under Poisson model) as*

$$m_{\mathrm{hist}}^{\mathrm{poi}}(n, k, \varepsilon, \nu) := \min\left\{m \geq 0 : \exists \hat{I} \text{ such that } \forall P \in \tilde{\mathcal{P}}^n\right.$$

$$\left.\Pr[\hat{I} = 1, \mathbb{1}\{\inf_{P' \in \tilde{\mathcal{H}}_k^n(\nu)} \mathrm{TV}(P', P) \geq \varepsilon(1 + \nu)\}] + \Pr[\hat{I} = 0, \mathbb{1}\{P \in \tilde{\mathcal{H}}_k^n(\nu)\}] \leq 1/10\right\},$$

where $\hat{I}$ is a binary indicator measurable with respect to $M = (M_1, M_2, \dots) \sim \mathrm{Poi}(mP_i)$.[7]

## B.1 Construction of Moment-Matching Random Variables

**Fact 5.** *Suppose $p$ is a degree-$d$ polynomial with distinct roots $r_1, \cdots r_d$. Then, for every $0 \leq k \leq d-2$, $\sum_{i=1}^{d} \frac{r_i^k}{p'(r_i)} = 0$.*

*Proof of Lemma 6.* We construct such a pair explicitly, relying on properties of Chebyshev polynomials. For every integer $d \geq 0$, recall that the corresponding Chebyshev polynomial (of the first kind) $T_d$ is given by $T_d(x) = \cos(d \cdot \arccos(x))$ for $x \in [-1, 1]$. (and can be shown to be a polynomial of degree $d$). Then, let $\Delta := \frac{\sqrt{kn}}{C \cdot \log^2 n}$ and $d := c \cdot \log n$ (where $c, C > 0$ are two absolute constants, suitably large), and consider the polynomial

$$p(x) = x \left( x - \frac{1}{n} \right) \left( x - \frac{2}{n} \right) T_d \left( 1 - \Delta \cdot x \right), \tag{16}$$

for $x \in [0, 1/\Delta]$. This is a degree-$(d+3)$ polynomial, whose roots are listed in the table below along with the corresponding values of its derivative.

| Roots | $p'$ |
|---|---|
| $r_0 = 0$ | $\frac{2}{n^2} \cdot T_d(1) = \frac{2}{n^2}$ |
| $r_1 = \frac{1}{n}$ | $-\frac{1}{n^2} \cdot T_d\left(1 - \frac{\Delta}{n}\right)$ |
| $r_2 = \frac{2}{n}$ | $\frac{2}{n^2} \cdot T_d\left(1 - \frac{2\Delta}{n}\right)$ |
| $r_{2+m} = \frac{1}{\Delta}\left(1 - \cos\left(\frac{2m-1}{2d}\pi\right)\right)$ for $1 \leq m \leq d$ | $|p'(r_{2+m})| = \Theta\left(\frac{m^5}{\Delta^2 d^4}\right)$ for $1 \leq m \leq d$. |

The roots in the last row are those associated with the Chebyshev Polynomial $T_d$, and the last line relies on the facts that, for $r$ such that $T(1 - \Delta r) = 0$, we have

$$p'(r) = -\Delta r(r - 1/n)(r - 2/n)T_d'(1 - \Delta r)$$

and that $T_d'(\cos \theta) = \frac{d \sin(d\theta)}{\sin \theta}$. This latter identity implies that $|T_d'(1 - \Delta r_{2+m})| = \frac{d}{|\sin \frac{2m-1}{2d}|} = \Theta\left(\frac{d^2}{m}\right)$, and since $r_{2+m} = \Theta(\frac{m^2}{\Delta d^2})$ we get the claimed bound:

$$|p'(r_{2+m})| = \Theta\left( \Delta \cdot \frac{m^2}{\Delta d^2} \cdot \max\left(\frac{m^2}{\Delta d^2}, \frac{1}{n}\right)^2 \cdot \frac{d^2}{m} \right) = \Theta\left(\frac{m^5}{\Delta^2 d^4}\right) \tag{17}$$

Some of these $d + 3$ roots have positive derivatives, while others have negative derivatives: this will tell us which ones to use for our construction of $U$, and which ones for $U'$. Namely, for root $r$, we set (1) $\Pr(U = r) = 0$ and $\Pr(U' = r) \propto \frac{1}{p'(r)}$ if $p'(r) > 0$, and (2) $\Pr(U = r) \propto \frac{1}{p'(r)}$ and $\Pr(U' = r) = 0$ otherwise.

We now derive some bounds on the derivatives. Firstly, for sufficiently large choice of $C$ (compared to $c$), notice that the weights on the Chebyshev polynomial's roots are overall bounded by

$$\sum_{m=1}^{d} \frac{1}{p'(r_{m+2})} = \Theta(1) \cdot \Delta^2 d^4 \sum_{m=1}^{d} \frac{1}{m^5} \ll nk, \tag{18}$$

since $\Delta^2 d^4 = kn \cdot (c^2/C)^2$ and the series $\sum_m \frac{1}{m^5}$ is convergent. Secondly, we claim that

$$T_d\left(1 - \Delta/n\right), T_d\left(1 - 2\Delta/n\right) \geq \cos(\pi/3) = \frac{1}{2}. \tag{19}$$

---

[7] We remark that the choice of the constant $1/10$ for error rate is arbitrary and our argument can easily be adapted to show a lower bound on the sample complexity under any constant error rate $\delta \in (0, 1/2)$.

This is because $\cos(\pi/3d) \le 1 - \frac{2}{d^2}$; while $1 - \Delta/n \ge 1 - \frac{1}{C \log^2 n} \ge 1 - \frac{2}{d^2}$, the latter inequality again for a sufficiently large choice of $C$ (with respect to $c$). The claim then follows by noticing that $T_d$ is monotonically increasing in the region $[\pi/3d, 1]$ and $T_d(\cos(\pi/3d)) = \cos(\pi/3)$. We then proceed to verify each property claimed in the lemma.

1. By our construction, $U$ is only supported on roots with negative derivatives. Hence, it can only takes values from $r_1 = \frac{1}{n}$ and some of the roots of the Chebyshev polynomials $r_{2+m}$. Moreover, by Equations (18), (19), the probabilities are bounded respectively by (before renormalization) $\Pr[U = \frac{1}{n}] \propto \Omega(n^2)$ and $\Pr[U \ne \frac{1}{n}] \propto \sum_{m=1}^{d} \mathbb{1}\{p'(r_{2+m}) < 0\}|p'(r_{m+2})| \ll nk$. It then follows that $\Pr[U \ne \frac{1}{n}] \ll \frac{nk}{nk+n^2} \le \frac{k}{n}$.

2. Similarly, we have $\Pr[U' = 0], \Pr[U' = 2/n] \propto \Omega(n^2)$ and by (19) are within a factor 2 of each other. Combined with the fact that

$$\Pr[U' \notin \{0, 2/n\}] \propto \sum_{m=1}^{d} \mathbb{1}\{p'(r_{2+m}) > 0\}p'(r_{2+m}) \ll nk\,,$$

this immediately yields after renormalization that $\Pr[U' = 0], \Pr[U' = 2/n] \ge \frac{1}{3}$.

3. The largest values $U, U'$ can take are the largest root of the Chebyshev polynomial, which is at most $\max_{m \in [d]} \frac{1}{\Delta}\left(1 - \cos\left(\frac{2m-1}{2d}\pi\right)\right) \le \frac{2}{\Delta} = \frac{2C \log^2 n}{\sqrt{kn}}$.

4. First, as we used earlier, recall that by Taylor expansion of cos

$$r_{m+2} = \frac{1}{\Delta}\left(1 - \cos\left(\frac{2m-1}{2d}\pi\right)\right) = \Theta\left(\frac{m^2}{\Delta d^2}\right)$$

Then, it holds

$$\mathbf{E}[U] = \Pr\left[U = \frac{1}{n}\right] \cdot \frac{1}{n} + \sum_{m=1}^{d} \Pr[U = r_{m+2}] \cdot r_{m+2}$$

$$\le \frac{1}{n} + \Theta(1) \cdot \sum_{m=1}^{d} \frac{\Delta^2 d^4}{n^2 m^5} \cdot \frac{m^2}{\Delta d^2}$$

$$\le \frac{1}{n} + \Theta\left(\frac{\Delta d^2}{n^2}\right) \cdot \sum_{m=1}^{d} \frac{1}{m^3} \le \frac{1}{n}\left(1 + O(\sqrt{k/n})\right).$$

The argument for $\mathbf{E}[U'] = \frac{1}{n}(1 + O(\sqrt{k/n}))$ is similar.

5. The claim follows from Fact 5.

$\square$

## B.2 Lower Bounds under Poisson Sampling

*Proof of Proposition 2.* Our goal is to argue $H$ and $H'$, specified in Equation (20) below, satisfy the following properties (i) $H$ and $H'$ are positive measures with total mass $1 + \Theta(\varepsilon)$ with probability at least $99/100$; (ii) $H$ is a $k$-histogram with probability at least $99/100$ and $H'$ is $\Omega(\varepsilon)$-far away from any $k$-histogram with probability at least $99/100$; and (iii) the distributions of the vectors $M = \{M_1, M_2, \cdots M_n\}$ and $M' = \{M'_1, M'_2, \cdots M'_n\}$, where $M_i \sim \mathrm{Poi}(m \cdot H_i)$ and $M'_i \sim \mathrm{Poi}(m \cdot H'_i)$, are $1/4$-close to each other in TV distance as long as $m = o(\sqrt{kn}/\log^2 n)$. If all these properties are satisfies, the result follows by applying Le Cam's Lemma. To do so, we define the measures $H$, $H'$ as

$$H = \left(\frac{1}{n} + \varepsilon U^{(1)}, \cdots, \frac{1}{n} + \varepsilon U^{(n)}\right) \quad H' = \left(\frac{1}{n} + \varepsilon U'^{(1)}, \cdots, \frac{1}{n} + \varepsilon U'^{(n)}\right) \qquad (20)$$

where $U^{(1)}, \cdots, U^{(n)}$ and $U'^{(1)}, \cdots U'^{(n)}$ are $n$ i.i.d. copies of the random variables $U, U'$ defined in Lemma 6. We proceed to verify that each property in our goal is satisfied.

(i) We first observe that the mass of $H$ is simply $1 + \varepsilon \sum_{i=1}^{n} U^{(i)}$. As stated in Lemma 6, we have $\mathbf{E}\left[U\right] = \frac{1}{n}\left(1 + O(\sqrt{k/n})\right)$. Since $U^{(i)}$s are just i.i.d. copies of $U$, this further implies that $\mathbf{E}\left[\sum_{i=1}^{n} U^{(i)}\right] = 1 + O(\sqrt{k/n}) = O(1)$. Hence, by Markov's inequality, we have $\sum_{i=1}^{n} U^i = \Theta(1)$ with probability at least $99/100$. Similar arguments hold for $H'$. This shows claim (i).

(ii) Turning to (ii), recall that by construction of $U$ we have $\Pr\left[U \neq \frac{1}{n}\right] \ll k/n$. Hence, with probability $99/100$, there are at most $(k-1)/2$ entries in $H$ with mass other than $1/n$, which makes $H$ a $k$-histogram. To argue the second part, i.e., that $H'$ is far away from any $k$-histogram, we first lower bound the number of adjacent pairs $(i, i+1)$ such that $H_i = 0, H_{i+1} = \frac{2}{n}$. We call such an adjacent pair a "right border pair." For $U'$, we have $\Pr\left[U' = 0\right], \Pr\left[U' = \frac{2}{n}\right] > 1/3$. Hence, in expectation, there are at least $\frac{1}{9}(n-1)$ such right border pairs. On the other hand, the variance of the number of right border pairs is at most $n$. Therefore, by Chebyshev's inequality (and assuming $n$ large enough), with probability $99/100$ there will be at least $n/100$ right border pairs. This implies that $H'$ is at least $\left(\frac{n}{100} - k\right) \cdot \frac{\varepsilon}{n} = \Omega(\varepsilon)$ far from a $k$-histogram. This concludes claim (ii).

(iii) Let $P_M$ and $P_{M'}$ be the distributions of the tuple of $m$ samples seen by the algorithm. By subadditivity of total variation distance, we have

$$\mathrm{TV}(P_M, P'_M) \leq n \cdot \mathrm{TV}\left(\mathbf{E}_U\left[\mathrm{Poi}\left(\frac{m}{n} + \varepsilon m U\right)\right], \mathbf{E}_{U'}\left[\mathrm{Poi}\left(\frac{m}{n} + \varepsilon m U'\right)\right]\right). \quad (21)$$

To handle the right-hand-side, we will use the following lemma about distance between mixtures of Poisson distributions.

**Lemma 8** ([WY+19, Lemma 4]). *Let $V, V'$ be random variables taking values in $[0, \Lambda]$. If $\mathbf{E}\left[V^j\right] = \mathbf{E}\left[V'^j\right]$ for $1 \leq j \leq L$, then*

$$\mathrm{TV}(\mathbf{E}\left[\mathrm{Poi}(V)\right], \mathbf{E}\left[\mathrm{Poi}(V')\right]) \leq \left(\frac{e\Lambda}{2L}\right)^L.$$

Now, let $\Lambda := m\left(\frac{1}{n} + c'\varepsilon\frac{\log^2 n}{\sqrt{kn}}\right)$ and $L := c\log n$, where $c, c' > 0$ are as in Lemma 6. It is straightforward to check from Lemma 6 that the random variables $V := \frac{m}{n} + \varepsilon m U$, $V' := \frac{m}{n} + \varepsilon m U'$ satisfy the assumptions of Lemma 8. This implies the existence of a (small) absolute constant $c'' > 0$ such that, if $m \leq c'' \min\left(n \log n, \frac{\sqrt{kn}}{\varepsilon \log n}\right)$, then by (21)

$$\mathrm{TV}(P_M, P'_M) \leq n \cdot \left(\frac{e}{2c} \cdot m\left(\frac{1}{n \log n} + c'\frac{\varepsilon \log n}{\sqrt{kn}}\right)\right)^{c\log n} \leq n \cdot \frac{1}{n^2} = \frac{1}{n} < \frac{1}{4}. \quad (22)$$

Combining Equations (22) and (21) then finishes the proof of claim (iii).

Since $H$ and $H'$ satisfy all the properties listed, by Le Cam's Lemma, no algorithm can distinguish between the two distributions with probability more than $3/4 + 2/100 \leq 9/10$ when

$$m = o\left(\min\left(n \log n, \frac{\sqrt{kn}}{\varepsilon \log n}\right)\right)$$

under the Poissonization model. Notice that $\min\left(n \log n, \frac{\sqrt{kn}}{\varepsilon \log n}\right)$ is exactly $\frac{\sqrt{kn}}{\varepsilon \log n}$ under the assumption $\varepsilon \cdot \log^2 n \geq \sqrt{k/n}$. This concludes the proof. $\qquad \square$

*Proof of the second part of Proposition 2.* We already showed above, while establishing the first part of Proposition 2, that, with $99\%$ probability, the total masses of $H, H'$ are $1 + \Theta(\varepsilon)$, and that $H$ is a $k$-Histogram Approximate Probability Vector while $H'$ is $\Omega(\varepsilon)$ far away any $k$-Histogram Approximate Probability Vector. It thus only remains to show the indistinguishability. To do so, let as before $P_M$ and $P_{M'}$ denote the distributions of the tuple of $m$ samples seen by the algorithm: we

then need to establish that $P_M$ and $P_{M'}$ are within total variation distance $1/4$ when $m = \tilde{o}(k/\varepsilon^2)$, for which is suffices to bound by $1/4$ the RHS below:

$$\text{TV}\left(P_M, P'_M\right) \le n \cdot \text{TV}\left(\mathbf{E}\left[\text{Poi}\left(\frac{m}{n} + \varepsilon m U\right)\right], \mathbf{E}\left[\text{Poi}\left(\frac{m}{n} + \varepsilon m U'\right)\right]\right). \tag{23}$$

We will now use a result from [Han19], restated below:

**Theorem 6** (Theorem 4 from [Han19]). *For any $\Lambda > 0$ and random variables $X, X'$ supported on $[-\Lambda, \infty)$, we have*

$$\text{TV}(\mathbf{E}[\text{Poi}(\Lambda + X)], \mathbf{E}[\text{Poi}(\Lambda + X')]) \le \frac{1}{2}\left(\sum_{\ell=0}^{\infty} \frac{\left|\mathbf{E}[X^\ell] - \mathbf{E}[X'^\ell]\right|^2}{\ell! \Lambda^\ell}\right)^{1/2}.$$

Recall that the first few moments of $U, U'$ are identical i.e., $\mathbf{E}[U^t] = \mathbf{E}[U'^t]$ for $1 \le t \le L := c \cdot \log n$. Let $X := \varepsilon m U$, $X' := \varepsilon m U'$, and $\Lambda := \frac{m}{n}$. Notice that we indeed have $|X|, |X'| \le \Lambda$ under the assumption $\varepsilon \ll \sqrt{k/n}/\log^2 n$ since

$$\max(|X|, |X'|) = \varepsilon \cdot m \cdot \max(|U|, |U'|) \le \varepsilon \cdot m \cdot O\left(\frac{\log^2 n}{\sqrt{kn}}\right) \le m/n = \Lambda.$$

Applying Theorem 6 then gives

$$4 \cdot \text{TV}\left(\mathbf{E}\left[\text{Poi}\left(\frac{m}{n} + \varepsilon m U\right)\right], \mathbf{E}\left[\text{Poi}\left(\frac{m}{n} + \varepsilon m U'\right)\right]\right)^2$$

$$\le \sum_{\ell=0}^{\infty} (\varepsilon m)^{2\ell} \frac{\left|\mathbf{E}[U^\ell] - \mathbf{E}[U'^\ell]\right|^2}{\ell!(m/n)^\ell}$$

$$= \sum_{\ell=L+1}^{\infty} \left(\varepsilon^2 mn\right)^\ell \frac{\left|\mathbf{E}[U^\ell] - \mathbf{E}[U'^\ell]\right|^2}{\ell!} \qquad \text{(the first } L \text{ moments match)}$$

$$\le \sum_{\ell=L+1}^{\infty} \left(\varepsilon^2 mn\right)^\ell \frac{1}{\ell!} \left(\frac{c' \log^2 n}{\sqrt{kn}}\right)^{2\ell} \qquad (|U|, |U'| \le \frac{c' \log^2 n}{\sqrt{nk}} \text{ (Lemma 6))}$$

$$= \sum_{\ell=L+1}^{\infty} \left(c'^2 \cdot \log^4 n \cdot \varepsilon^2/k \cdot m\right)^\ell \frac{1}{\ell!}.$$

Set for convenience $\kappa := c'^2 \cdot \log^4 n \cdot \varepsilon^2/k \cdot m$. Notice that $\kappa \ll L$ when $m = o\left(\frac{k}{\varepsilon^2 \log^3 n}\right)$. Denoting by $Y$ a $\text{Poi}(\kappa)$ random variable, this leads to

$$4\text{TV}\left(\mathbf{E}\left[\text{Poi}\left(\frac{m}{n} + \varepsilon m U\right)\right], \mathbf{E}\left[\text{Poi}\left(\frac{m}{n} + \varepsilon m U'\right)\right]\right)^2 \le \sum_{\ell=L+1}^{\infty} \frac{\kappa^\ell}{\ell!} = e^\kappa \Pr[Y \ge L+1]$$

$$\le e^\kappa e^{-\frac{(L+1-\kappa)^2}{2(L+1)}} = e^{-\frac{1}{2}\left(L+1+\frac{\kappa^2}{L+1}\right)} \le e^{-\frac{L}{2}},$$

where the second inequality is by standard Poisson concentration (see, e.g., the note [Can]) and $\mathbf{E}[Y] = \kappa \ll L$. This immediately gives

$$\text{TV}\left(\mathbf{E}\left[\text{Poi}\left(\frac{m}{n} + \varepsilon m U\right)\right], \mathbf{E}\left[\text{Poi}\left(\frac{m}{n} + \varepsilon m U'\right)\right]\right) \le \frac{1}{2} e^{-L/4} \tag{24}$$

Combining Equations (23) and (24) then yields

$$\text{TV}\left(P_N, P'_N\right) \le n \cdot e^{-L/4} = n \cdot e^{-c \log n/4} < 1/4.$$

for sufficiently large $c$. By Le Cam's Lemma, no algorithm can distinguish between the two distributions as constructed in Equation (20) with probability more than $3/4 + 2/100 < 9/10$ when $m = o\left(\frac{k}{\varepsilon^2 \log^3 n}\right)$ under the Poissonization model. $\qquad \square$

## B.3 Proof of Lower Bound in Theorem 1

*Proof of Lower Bound Part of Theorem 1.* By Proposition 2, it holds that $m_{\text{hist}}^{\text{poi}} \geq \Omega(k/(\varepsilon^2 \cdot \log^3 k) + \sqrt{kn}/(\varepsilon \cdot \log k))$. We proceed to argue for sample complexity lower bound under standard sampling. Suppose we are given a tester for fixed sample size such that it succeeds with high probability as long as it is given more than $m^*$ samples. Assume that we want to use it to test whether an unknown measure $P$ is a $k$-histogram under Poisson sampling model with $\tilde{m} \sim \text{Poi}(m_{\text{hist}}^{\text{poi}}(n, \ell, \varepsilon) \|P\|_1)$ samples. We can construct an estimator which invokes the fixed sample size tester whenever $\tilde{m} \geq m^*$ and outputs fail otherwise.

By our lower bound result for the Poisson sampling model, the estimator fails with probability at least $1/10$. On the other hand, the estimator based on the fixed-sample tester succeeds with high probability whenever $\tilde{m} > m^*$. Together this implies that

$$m^* \geq (1-\nu) \cdot m \cdot \left(1 - O\left((1-\nu)^{-1} \cdot m^{-\frac{1}{2}}\right)\right),$$

where $m := m_{\text{hist}}^{\text{poi}}(n, \ell, \varepsilon, \nu)$. Since $\nu < 1$, it then holds

$$m^* > \Omega(1) \cdot m_{\text{hist}}^{\text{poi}}(n, \ell, \varepsilon, \nu) \geq \Omega(k/(\varepsilon^2 \cdot \log^3 k) + \sqrt{kn}/(\varepsilon \cdot \log k)).$$

Finally, we remark that the standard lower bound construction and analysis for uniformity testing can be shown to still apply to testing $k$-histograms (see [Can16, Proposition 4.1]), making formal the intuitive statement that testing whether a distribution is a $k$-histogram is at least as hard as testing whether a distribution is uniform. This shows that we also have $m^* \geq \Omega(\sqrt{n}/\varepsilon^2)$, and concludes the proof. $\qquad\square$

## C Application to Model Selection

In this section, we detail how our testing algorithm, by a standard reduction, can be used for *model selection*, i.e., to select a suitable parameter $k$ in order to succinctly represent the data.

**Theorem 7.** *There exists an algorithm which, given samples from an unknown distribution $\mathbf{p}$ on $[n]$, an error parameter $\varepsilon \in (0,1]$, and a failure probability $\delta \in (0,1]$, outputs a parameter $1 \leq K \leq n$ such that the following holds. Denote by $1 \leq k \leq n$ the smallest integer such that $\mathbf{p} \in \mathcal{H}_k^n$. With probability at least $1-\delta$, (1) the algorithm takes $\widetilde{O}((\sqrt{n}+k)/\varepsilon^2 \cdot \log(1/\delta))$ samples, (2) $1 \leq K \leq 2k$, and (3) $\mathbf{p}$ is at TV distance at most $\varepsilon$ from some $H \in \mathcal{H}_K^n$. Moreover, the algorithm runs in time $\text{poly}(m)$, where $m$ is the number of samples taken.*

Before proving the theorem, we discuss the various aspects of its statement. (1) guarantees that, with high probability, the algorithm does not take more samples than what it would if it were *given $k$* and just needed to test whether it was the right value. Moreover, as most tasks on $k$-histograms require at least a number of samples growing linearly in $k$, this guarantees that whenever $k \gg \sqrt{n}$ the cost of the model selection step is negligible in front of all subsequent tasks. Item (2), ensures that the output of the algorithm is a good approximation of the true, *optimal* value $k$; that is, that the model selected is essentially as succinct as it gets. Finally, (3) guarantees that even when the output is such that $K \ll k$, the parameter $K$ is still good: that is, approximating $\mathbf{p}$ by a $K$-histogram still leads to a sufficiently accurate representation (even though it is even more succinct that what the *true* parameter $k$ would yield).

*Proof.* The model selection algorithm is quite simple, and works by maintaining a current "guess" $K$ as part of a doubling search, and using the tester of to check if the current value is good. Specifically: for $0 \leq j \leq \lceil \log_2 n \rceil$, run the testing algorithm of Theorem 1 (**Divide-And-Learn-And-Sieve**) with parameters $n, \varepsilon, 2^j$, and $\delta_j := \frac{\delta}{2(j+1)^2}$. If the testing algorithm returns Accept, then return $K := 2^j$ and stop; otherwise, continue the loop.

Let $L \leq \lceil \log_2 n \rceil$ be the number of iterations before the algorithm stops. By a union bound, the probability that all $L$ invocations of the testing algorithm behaved as they should is, by a union bound, at least

$$1 - \sum_{j=0}^{L} \delta_j = 1 - \sum_{j=0}^{L} \frac{\delta}{2(j+1)^2} \geq 1 - \sum_{j=0}^{\infty} \frac{\delta}{2(j+1)^2} \geq 1 - \delta.$$

Condition on this being the case. Since the algorithm, as soon as $j \geq \lceil \log_2 k \rceil$, returns Accept (since then $\mathbf{p} \in \mathcal{H}_{2^j}^n$), we get that $L := \lceil \log_2 k \rceil$, and that the output satisfies $K \leq 2k$, giving (2). Moreover, if $K < k$, then by the guarantee of the testing algorithm this means that $\mathbf{p}$ was *not* $\varepsilon$-far from $\mathcal{H}_K^n$, showing (3). Finally, the sample complexity is then

$$\sum_{j=0}^{L} \tilde{O}\left( \frac{\sqrt{n}}{\varepsilon^2} \log \frac{1}{\delta_j} + \frac{2^j}{\varepsilon^2} \log \frac{1}{\delta_j} \right) = \tilde{O}\left( \frac{\sqrt{n}}{\varepsilon^2} + \frac{k}{\varepsilon^2} \right) \log \frac{1}{\delta}$$

since $L = O(\log k)$. This shows (1), and concludes the proof. $\qquad\square$

As a final remark, we note that the guarantee provided in (2) can be improved to the optimal $1 \leq K \leq k$, by modifying slightly the above procedure. Namely, after finding some $K$ such that $1 \leq K \leq 2k$ as before, one can run the testing algorithm for $K/2 \leq i \leq K$ (not a binary search anymore), each time with parameters $n, \varepsilon, i$, and $\delta_i = \delta/K$. By a union bound, this incurs an extra $\delta$ probability of failure, and an additional $\tilde{O}\big((\sqrt{n} + k)/\varepsilon^2 \cdot \log(1/\delta)\big)$ samples overall, but now the output after this second step will be guaranteed to be at most $k$ (with high probability).