# OpenReview forum: "Nearly-Tight Bounds for Testing Histogram Distributions"
_NeurIPS.cc/2022/Conference — NeurIPS 2022 Accept_

### Official Review · Reviewer_YQdn · 2022-06-25

**Rating:** 6
**Confidence:** 3
**Soundness:** 3 good
**Presentation:** 3 good
**Contribution:** 3 good

**Summary:**

The paper, as suggested in the title, provides a new algorithm for testing whether a given distribution is close to a k-histogram, where the measure of performance is the sample complexity, and provides a nearly matching lower bound.

**Questions:**


Have you implemented the algorithm?  i ask as you make claims to practical relevance, but also for these complex analyses I might suggest that an implementation is a good check that everything works (as subtle bugs in theoretical analyses are hard to find).

I couldn't help but wonder about a related problem, focused on what is the importance here of the data being a histogram, with intervals having the same value.  What if instead it was just that p_i was supposed to be close to one of k values for all i, but they need not be arranged in intervals?  (That is, the data is a permutation of or could be permuted into a distribution that is close to a k-histogram.)  Is this problem known/understood?  What is the importance of "intervals" in the sample complexity here?

**Strengths And Weaknesses:**

Strengths:

Clear formal problem.
Proofs/arguments seem difficult and of high quality.

Weaknesses:
This is a theoretical paper.  [It is my opinion that it would be "more at home" at a theoretical conference, ranging from COLT or RANDOM to SODA/FOCS/STOC, but I will not focus on scope in my evaluation.]
No simulations to verify the work or compare with other approaches -- purely a theoretical result.
Ties to learning generally are weak (a paragraph devoted to model selection, that's about it).
Small number of typos.

Originality:  Appears to be original theoretical results.
Significance:  Appears to be a significant advance on this problem.
Quality:  Challenging theorems.
Clarify:  Definitely hard to follow and dense for someone not in the area, as one might expect for a theory paper.
Significance:  Worthwhile problem in something of a niche area.

I think in this sort of paper it is worthwhile for the authors to make clear what their goal is.  Here their goal is apparently to get near-optimal upper and lower bounds.  But are these algorithms practical?   Are they actually better than the prior work they cite in a practical sense?  As a theoretical paper, the authors don't seem to care to address these issues, but since their motivation (for this conference at least) is that this may be useful for model selection, it does seem to me that some experimentation or even just showing they have implemented the algorithm would have been useful.

---

> ### Author Response · Authors · 2022-08-02
> **About testing histogram up to permutation**
>
> We appreciate the reviewer’s concerns regarding practicality, and refer to the response to reviewer #VmGT for this aspect. Regarding the second question, i.e., the (related) problem of testing whether a distribution is a k-histogram up to a permutation, we are unaware of any previous work on this question. More specifically, to the best of our knowledge, this has not been studied before in the distribution testing literature as long as $k\geq 2$ (for $k=1$, this boils down to the standard task of testing uniformity). We also note that our lower bound does apply to this question, as our family of “hard instances” is invariant by permutation and each “yes” distribution takes O(k) different values. However, besides this remark, it is worth noting that the problem suggested by the reviewer is orthogonal to the one addressed in our paper, and we suspect would require a paper in its own right, with different techniques and results. We would be happy to discuss this in more detail, if the reviewer has any additional questions or remarks.

---

### Official Review · Reviewer_VmGT · 2022-07-11

**Rating:** 8
**Confidence:** 4
**Soundness:** 3 good
**Presentation:** 4 excellent
**Contribution:** 3 good

**Summary:**

Distribution testing is a field of sublinear algorithms that aims to certify that a input probability distribution has some property or is $\epsilon$-far from it, e.g., in total variation distance. One such property is whether a distribution on $[n]$ is a $k$-histogram, i.e., the distribution's support can be partitioned into $k$ continuous intervals so that on each interval, the distribution is uniform. The so far best algorithm draws $\tilde{O}(\sqrt{kn} / \epsilon^3)$ samples from the input, and a lower bound of $\tilde{\Omega}(\sqrt{n} / \epsilon^2 + k / \epsilon)$ is known. If $k$ or $\epsilon$ are a function of $n$ (i.e., only mild bucketing or high precision), these bounds are not tight. In this submission, the authors present an algorithm with complexity $\tilde{O}(\sqrt{nk} / \epsilon + k / \epsilon^2 + \sqrt{n} / \epsilon^2)$, and a matching lower bound (up to polylogarithmic factors).

The algorithm designed by the authors partially picks up a previous (and failed) attempt from related work and tests the distribution learning it first. In particular, the algorithm tries to learn a $k$-histogram model from the input and detects breaks between intervals that (are likely) uniformly distributed. The algorithms is adaptive, and iterates on these breaks to either reduce the uncertainty sufficiently or certify that the input is far from a $k$-histogram. The lower bound is based on a classic information-theoretic scheme, where the challenge lies in the analysis of the specific instance.

Remark on the confidence of the rating: I didn't check all the math carefully. The arguments are plausible, but the whole proof (main part + appendix) that would need to be checked is too much.

**Questions:**

-

**Limitations:**

-

**Strengths And Weaknesses:**

Strengths:

* Upper and lower bounds matching in $n,k,\epsilon$ up to polylogarithmic factors, which is quite strong as a as statement.
* The theoretical analysis is sophisticated and highly non-trivial.

Weaknesses:

* For a proof of practical implications, it would be good to either have explicit bounds (with constants spelled out, no asymptotics) or experiments.

---

> ### Author Response · Authors · 2022-08-02
> **About practical application**
>
> This is a good point regarding practical implications, and we agree
> that an implementation, complemented with an extensive experimental
> evaluation of our algorithm is the next natural step. We however
> highlight that the main goal of the current paper is to understand the
> fundamental sample complexity of histogram testing – and indeed,
> theoretical findings and results are fully in scope at NeurIPS (cf.
> the Theory section of the CfP). Implementing and evaluating our
> algorithm is thus beyond the scope of the current paper, and we plan
> on (and have started) addressing this in future work.
>
>  As a side remark, we did not try to optimize the constants in the
> proofs, as this would have made the arguments longer and more
> technical, and would have distracted from the main results (and,
> further, our work relies on some lemmas from previous work where those
> constants were either unspecified or unnecessarily large to begin
> with).

---

### Official Review · Reviewer_oQwe · 2022-07-13

**Rating:** 7
**Confidence:** 3
**Soundness:** 4 excellent
**Presentation:** 4 excellent
**Contribution:** 3 good

**Summary:**

This paper falls in the topic of testing properties of discrete distributions using *sublinear* number of independent samples. The testing problem considered is the following. Given independent samples from an unknown distribution $D$ over sample space $[n]$, an integer $k \leq n$, and an error parameter $\varepsilon$. Decide whether $D$ is a $k$*-histogram* over $[n]$ or $D$ is $\varepsilon$-far (in total variation distance) from every $k$-histogram over $[n]$. A $k$-histogram is a distribution that is piecewise uniform over $k$ intervals over $[n]$. The goal is to design an efficient algorithm  that uses as little samples as possible,  and also to establish lower bounds on the number of samples needed.

The main contribution of the  paper is an efficient algorithm with near-optimal sample complexity for this testing problem with a matching lower bound (up to polylog factors). In particular they (1) design a time efficient algorithm that uses $\tilde{O}(\sqrt{nk}/\varepsilon+k/\varepsilon^2+\sqrt{n}/\varepsilon^2)$ samples, (2) establish a lower bound of  $\tilde{\Omega}(\sqrt{nk}/\varepsilon+k/\varepsilon^2+\sqrt{n}/\varepsilon^2)$ on the number of samples require.  This settles the sample complexity of the histogram-testing problem for all parameter regimes up to polylog factors.

**Questions:**

As mentioned, the limitation of the paper is its appeal to a wider audience. I think it is inherent in the problem considered. So it will be difficult, but it will be nice if the authors can motivate this problem for a wider audience.

**Limitations:**

I do not see any section addressing the limitations and potential negative societal impact of their work. However, for this work that may not be relevant.

**Strengths And Weaknesses:**

*Strengths:* Histogram testing problem has been of interest in the distribution testing literature for the past decade. This paper can be viewed roughly as a culmination of research on this problem, at least from a sample complexity perspective. While the basic approaches used are from earlier works, the paper is technically very strong. The paper is well written.

*Weaknesses:* The histogram testing problem considered in this paper may not appeal to a broader community. While technical contribution is strong, the paper does not introduce any new approaches/techniques that may be of wider applicability. Thus this is a niche paper.

Overall, this is a strong paper in the specific area of distribution testing and I like it.

---

> ### Author Response · Authors · 2022-08-02
> **About audience and motivation**
>
> We are thankful to the reviewer for their kind words. Regarding the motivation for a wider audience, we remark that the histogram testing problem is studied in both the statistical hypothesis and database community due to its practical applications. In particular, an algorithm for the testing problem can be used as a preliminary step to compress a large amount of data into histograms that have relatively concise representations; indeed, this was the original motivation for studying this question in the database community (see, for instance, [ILR12]). This can in turn be helpful to many downstream tasks such as data aggregation and visualization, and go beyond the database applications to, e.g., memory-limited or communication-limited settings. We will emphasize this point in the final version of the paper.

---

### Official Review · Reviewer_ua2e · 2022-07-26

**Rating:** 8
**Confidence:** 4
**Soundness:** 4 excellent
**Presentation:** 4 excellent
**Contribution:** 4 excellent

**Summary:**

This paper falls in the broad area of distribution property testing; the property studied in this paper being whether a given unknown discrete distribution over an alphabet of size $n$ is a $k$-histogram or is $\epsilon$ away from any $k$-histogram in TV distance. Previous works showed a $\tilde{O}(\frac{\sqrt{kn}}{\epsilon^3})$ upper bound and a $\tilde{\Omega}(\frac{\sqrt{n}}{\epsilon^2} + \frac{k}{\epsilon})$ lower bound on the sample complexity for constant probability of error. This work characterizes optimal dependence of the sample complexity on $n,k,\epsilon$ up to log factors.

The tester largely builds on a previous work that had a flaw in its analysis. Specifically, at a high level, the tester in this paper learns an approximate measure that is close if the unknown distribution is a $k$-histogram. This approximate measure is learnt by partitioning the domain in $K$ intervals, constructing a "histogram-measure" on these intervals and then discarding 'bad intervals' that contribute to a large $\chi^2$ divergence. The main innovation lies in the last part where the tester removes fewer 'bad' intervals.

The lower bound relies on constructing hard instances using moment-matching random variables that are supported on roots of an appropriately constructed Chebyshev polynomial.

**Questions:**

I am not entirely convinced how practical the current formulation of the testing problem is, especially for histograms. For example, I can imagine a practitioner not knowing the value of $k$ precisely. In that case does it make sense to test against being $\epsilon$-away from a $k$-histogram since potentially a $k+1$-histogram can be $\epsilon$-away from a $k$-histogram? I do understand that the current formulation is ubiquitous in distribution property testing literature and perhaps makes analysis easier. But perhaps some discussion about why this formulation makes sense might improve the paper slightly.

Minor comments
1) m in Lemma 1 is not defined.
2) Figure 1 is an interesting way to depict tradeoffs, but it's not polished. I'd suggest TeXing the legend and increasing size.

**Limitations:**

I am satisfied with how the authors have addressed impact.

**Strengths And Weaknesses:**

This paper settles the histogram testing problem which earlier had a polynomial gap between the upper bound and lower bound on the sample complexity and was first studied in the current framework about a decade ago. The techniques in the paper are interesting and novel, although the tester mostly builds on previous work. The paper is written clearly and flows well.

---

> ### Author Response · Authors · 2022-08-02
> **About Formulation**
>
> We thank the reviewer for their time and feedback, and will address the two points (typo and figure) made. We respond to their question on the formulation of the problem below.
>
> In the case when the practitioner does not know the value of k precisely, one can follow the approach of model selection (as described in Appendix C in the supplemental) to do a binary for the smallest k such that the underlying distribution is a k-histogram. As discussed in that appendix, the binary search adds only a log(n) factor to the overall sample complexity. More generally, there are known reductions and ways to use the property testing formulation as a blackbox to handle specific scenarios, depending on the exact formulation sought by the practitioner; we will elaborate on this and provide some pointers in the final version.
>
> Besides, we remark that though there are potentially other interesting formulations of the histogram testing problem, in this work we focus on the standard formulation of histogram testing set forth years ago by a line of prior works in statistical hypothesis testing, [ILR12], [CDGR18], [DK16].

---

> > ### Comment · Reviewer_ua2e · 2022-08-08
> > **Post-rebuttal comment**
> >
> > Thanks for your response. I have no further clarifications.

---

### Meta-Review · Area_Chair_zFqB · 2022-08-27

**Recommendation:** Accept
**Confidence:** Certain

**Metareview:**

Given samples from an unknown discrete distribution, the goal of the paper is to test if it is a histogram over k bins or epsilon far away from all such distributions in the total variation distance. Authors provide a computationally efficient algorithm and further show that the sample complexity is near optimal. The reviewers agree that the results are interesting and novel and I recommend acceptance. As reviewers remark, the paper can benefit by a discussion on the motivation of this problem formulation and the practicality of the proposed approach. I strongly encourage authors to add a discussion addressing these comments in the final version of the paper.

**Award:**

No

---

### Decision · Program_Chairs · 2022-09-14

Accept